# The Advance of Fusarium Wilt Tropical Race 4 in Musaceae of Latin America and the Caribbean: Current Situation

**DOI:** 10.3390/pathogens12020277

**Published:** 2023-02-08

**Authors:** Gustavo Martínez, Barlin O. Olivares, Juan Carlos Rey, Juan Rojas, Jaime Cardenas, Carlos Muentes, Carolina Dawson

**Affiliations:** 1Instituto Nacional de Investigaciones Agrícolas (INIA-CENIAP), Avenida Universidad vía El Limón, Maracay 02105, Venezuela; 2Grupo de Investigación en Gestión de la Biodiversidad, Campus Rabanales, Universidad de Córdoba, Carretera Nacional IV, km 396, 14014 Córdoba, Spain; 3Programa Nacional de Frutales, Instituto Nacional de Innovación Agraria (INIA), La Molina 15024, Peru; 4FAO Plant Protection International Consultant, Manizales 170004, Colombia; 5Agencia de Regulación y Control Fito y Zoosanitario (AGROCALIDAD), Quito 170516, Ecuador; 6Center for International Cooperation in Agricultural Research for Development (CIRAD), UPR GECO, F-34398 Montpellier, France; 7GECO, Univ Montpellier, CIRAD, TA B-26, 34398 Montpellier, France

**Keywords:** banana, *Musa* spp., soil pathogen, *Fusarium odoratissimum*, vascular wilt, quarantine, disease management

## Abstract

The fungus *Fusarium oxysporum* f. sp. *cubense* tropical race 4 (syn. *Fusarium odoratissimum*) (*Foc* TR4) causes vascular wilt in Musaceae plants and is considered the most lethal for these crops. In Latin America and the Caribbean (LAC), it was reported for the first time in Colombia (2019), later in Peru (2021), and recently declared in Venezuela (2023). This work aimed to analyze the evolution of *Foc* TR4 in Musaceae in LAC between 2018 and 2022. This perspective contains a selection of topics related to *Foc* TR4 in LAC that address and describe (i) the threat of *Foc* TR4 in LAC, (ii) a bibliometric analysis of the scientific production of *Foc* TR4 in LAC, (iii) the current situation of *Foc* TR4 in Colombia, Peru, and Venezuela, (iv) medium-term prospects in LAC member countries, and (v) export trade and local food security. In this study, the presence of *Foc* TR4 in Venezuela and the possible consequences of the production of Musaceae in the long term were reported for the first time. In conclusion, TR4 is a major threat to banana production in Latin America and the world, and it is important to take measures to control the spread of the fungus and minimize its impact on the banana industry. It is important to keep working on the control of *Foc* TR4, which requires the participation of the local and international industry, researchers, and consumers, among others, to prevent the disappearance of bananas.

## 1. Introduction

Edible Musaceae (bananas, plantains, and bluggoe) are considered basic foods for more than 400 million people worldwide due to their nutritional and economic contribution, which differentiates them from the rest of the fruits [1,2]. They are among the main important energy food items and the most exported fruits, with bananas being the most consumed and one of the main products that make up the daily movement in the international market [3], representing an essential source of income for thousands of rural households, particularly in Latin America and the Caribbean (LAC) [4,5,6,7].

For the year 2020, the global production was 163 Megaton (Mt) (74% bananas and the remaining 26% Musaceae), and a large part of this is used for self-consumption in many countries and the rest for export, where LAC contributes with the highest global proportion [8]. Initially, the international market was supported by the “Gros Michel” (GM) banana clone (*Musa* AAA, Gros Michel subgroup) until the end of the 1950s, when it succumbed to vascular wilt caused by the fungus *Fusarium oxysporum* f. sp. *cubense* (*Foc*) Race 1; it was necessary to replace it with natural clones resistant to that race, and they still predominate in current production systems [7,9,10].

Currently, the production of Musaceae is again threatened by a phytosanitary enemy that emerged from the past but was renewed [9]. This is the case for *Foc* Tropical Race 4 (TR4) (syn. *Fusarium odoratissimum*), with an intercontinental spread, which defines it as a pandemic, potentially threatening all Musaceae crops in LAC, since the official report of its presence in the American continent in 2019 [7,11,12,13].

Race 4 of the fungus poses significantly high risks to the world supply of Musaceae since it can directly affect the production of bananas and plantains. In *Foc* TR4-infected plants, rapid spread and infection can lead to a complete loss of yield [14]. An important characteristic of the fungus is represented by the longevity of the fungus in the soil; therefore, the lands infected by *Foc* TR4 cannot be used for plantations of Musaceae or other crops for decades, and this results in a drastic change in production, with new lands (which have not been intervened) as the only resource [11,15].

Depending on the disease severity, outbreaks can cause an increasing scarcity of soil free of this pathogen. In all the cases reported to date, once a plantation has been infected, the management of *Foc* TR4 has been extremely difficult and costly [16,17]. The impact of TR4 on the banana industry in LAC is significant [10,11,12,18]. The fungus has caused significant losses in productivity and profitability for banana growers in the region, which could hurt the livelihoods of small-scale farmers and the overall economy of the countries where banana is a major crop [19]. TR4 is considered a major threat to global banana production, as the fungus can spread easily through contaminated soil, water, and farm tools. The fungus can quickly spread to neighboring fields once it is established in a field, making it difficult to control [12,13,20].

To address the threat of TR4, measures such as quarantine, the destruction of infected plants, and the use of clean planting material were implemented by the governments of Latin American countries. Additionally, research to develop resistant varieties [21] and improve soil management practices is also ongoing [22]. Undoubtedly, this disease constitutes a real and particular threat to the livelihoods of small- and medium-sized banana producers in LAC, who mostly lack the economic means to develop measures against yield loss along with the increase in yield production costs [23,24,25].

The first reports of *Foc* TR4 in Colombia and Peru were in 2019 and 2021, respectively, and in the case of Venezuela, the recent declaration was issued in early 2023. Since then, the fungus has spread rapidly in these countries, causing significant losses in productivity and profitability for banana growers in the region [12,18]. The fungus has had a particularly severe impact on small-scale farmers, who are heavily dependent on banana production for their livelihoods [13].

Due to the serious repercussions for plantations infected by *Foc* TR4, there is often no accurate and complete information in countries such as Colombia, Peru, and Venezuela on the damage caused by *Foc* TR4. Although global estimates are not available, figures from some countries indicate that this disease has affected hundreds of ha in Colombia, Peru, and Venezuela [12,23]. In this regard, this study reports the updated numbers of banana areas affected by *Foc* TR4 in Colombia, Peru, and Venezuela and provides an analysis for the exchange of information and knowledge of the effects of *Foc* TR4 in LAC.

Numerous reviews of *Foc* TR4 have been published over the years, with the review by Pocasangre et al. [26] being the one of greatest interest in LAC for more than a decade. Therefore, our manuscript was based on a current perspective of the lethal disease of *Foc* TR4 for LAC. This work aimed to analyze the evolution of *Foc* TR4 in Musaceae in LAC between 2018 and 2022. This perspective contains a selection of topics related to *Foc* TR4 in LAC that address and describes (i) the threat of *Foc* TR4 in LAC, (ii) a bibliometric analysis of the scientific production on *Foc* TR4 in LAC, (iii) the current situation of *Foc* TR4 in Colombia, Peru, and Venezuela, (iv) medium-term prospects in LAC member countries, and (v) export trade and local food security. This research topic arose from the increasing diffusion and global impact of *Foc* TR4, which could affect a wide range of banana production systems in LAC.

## 2. Methodology

### 2.1. Type of Research and Method of Analysis

This study is a descriptive cross-sectional investigation considering the main components that characterize the reality of *Foc* TR4 in LAC during the period 2018–2022. Through this type of research, the inductive analysis method was used to characterize the specific situation of *Foc* TR4 and point out its characteristics and the actions that are being developed in this regard. Despite the amount of information available on this pathogen, not all of it has the necessary rigor, and for this reason, this manuscript collected reliable information provided by some Latin American experts with extensive experience on the subject. The inductive method allowed showing logical reasoning based on a series of observations of the reality of *Foc* TR4 in LAC that allow the production of protocols, regulations, action plans, and general conclusions. Likewise, the conception of this study can serve as a basis for research that requires a greater level of depth.

Analyzing the risk, consequences, and prospects for a region of the introduction and spread of a phytopathogenic organism is always a complicated task. From a scientific point of view, although analytical procedures and databases have improved remarkably, today, it is still difficult to predict the behavior of pathogens after they enter a new area. In 2014, the Food and Agriculture Organization of the United Nations (FAO) published the technical manual for the prevention and diagnosis of Fusarium Wilt, which established the framework for the risk analysis of *Foc* TR4. Currently, FAO [27] published the Regional Strategy for the Preparation, Prevention, Detection, Response, and Recovery of Latin America and the Caribbean to Fusarium Wilt of Musaceae, tropical race 4 (ER-TR4) and the Action Plan Regional (PAR-TR4), which served as a reference to analyze the issues discussed in this study.

### 2.2. Topic Analysis

This study includes five topics that aimed to offer an updated, realistic, practical, and critical review of the current state of *Foc* TR4 and the actions developed in Latin American territories where *Foc* TR4 is present. It consists of four general topics and one more specific on the situation and problems of *Foc* TR4 in Colombia, Peru, and Venezuela. Reading the different topics below leads to a deeper understanding of the complex system that constitutes *Foc* TR4 and its consequences for the production of Musaceae.

#### 2.2.1. The Threat of *Foc* TR4 in Banana Production

Data on the quantities of bananas harvested, produced, and yields published come from the FAO [8], which were compiled from country responses to the annual questionnaire of the FAO Intergovernmental Subgroup on Bananas; data from the United Nations Comtrade database [28] and official data from the National Statistical Offices and secondary data from documentary research were also used.

#### 2.2.2. Bibliometric Analysis of the Scientific Production on *Foc* TR4 in LA

In this study, the statistical method was applied that quantitatively analyzes the research works related to *Foc* in LAC through the mathematical method known as bibliometrics; for this, the VOSviewer software was used, and the methodology described in Van Eck and Waltman [29,30] was followed, which provided an overview of the quality of the studies, helped analyze the key areas of research, and helped predict the direction of future studies. To date, no bibliometric analysis of publications on *Foc* in LAC has been published. As this lethal disease in bananas has not been completely controlled and more knowledge must be obtained from these references, a bibliometric analysis of it is a critical need. Therefore, our study was performed in time to provide a broad understanding of *Foc* in LAC and future research directions.

The literature on *Foc* TR4 published between 2018 and 2022 related to LAC was extracted from the Web of Science (WOS) collection database. The following were used as keywords to reach the relevant publications: “*Fusarium oxysporum* f. sp. *cubense*”, “Fusarium wilt Musa”, and “*Foc* TR4”, delimiting the studies of Latin American and Caribbean countries. 

#### 2.2.3. Current Situation of *Foc* TR4 in Colombia, Peru, and Venezuela

Information was collected and provided by experts in plant health and Musaceae management in LAC. In addition, data sources and additional and complementary information were used, provided by the different National Phytosanitary Protection Organizations (NPPO) of the Colombian Agricultural Institute (ICA), National Agrarian Health Service of Peru (SENASA), and The National Institute of Comprehensive Agricultural Health (INSAI) in Venezuela, as they were the first countries with the presence of *Foc* TR4 in LAC, as well as various institutions and producers, providing valuable contributions.

#### 2.2.4. Medium-Term Prospects in LAC Member Countries

To develop the discussion in this section, the information provided by specialists in the management of these crops in LAC was used. Similarly, databases were used (FAO and national statistics from some countries in the region), and additional and complementary information was provided by the NPPO of several countries in the region, especially Colombia and Peru, as the first countries where the disease was reported, and others of particular interest, such as Ecuador for being the largest exporter of banana fruits in LAC and for sharing land borders with Colombia and Peru [31] and the case of Venezuela for its proximity to the Colombian Guajira [7,13].

#### 2.2.5. Export Trade and Local Food Security

Information about the impact of *Foc* TR4 in different countries was collected from previous studies published in scientific papers and FAO reports. In addition, banana export data come from national customs databases such as the Philippine Statistics Authority (PSA), general international databases (Trademap), and reports from the OCDE and FAO. Production surface figures come from official government bodies (Colombian Ministry of Agriculture) and INIA in Peru. World organic banana surfaces were retrieved from expert sources and CIRAD studies.

## 3. Results and Discussion

### 3.1. The Threat of Foc TR4 in Banana Production

For the year 2020, LAC contributed 24%, Africa 31%, and Asia 43% to the world production of edible Musaceae [8]. It is considered that the largest amount of fruit produced corresponds to bananas. The main world producer of bananas is India, with 31.5 million tons per year, followed by China and Indonesia. In Latin America, Ecuador is the main producer, with 6,745,688 tons per year, followed by Brazil, Colombia, Guatemala, Peru, Costa Rica, Mexico, the Dominican Republic, and Venezuela. These nine countries together account for more than half of the world’s production of Cavendish bananas (Figure 1b).

For the year 2019, the world export of bananas was 21 million tons, which represents an increase of 10.2% compared to 2018. It was made up of the contributions of LAC with 15.1, Asia with 5.1, and Africa with 0.8 million tons, equivalent to 74%, 22%, and 4%, respectively. LAC stands out with an increase of 3% compared to the previous year [2].

World banana production increased steadily between 1985 and 2019. Annual production increased by 49.0%, from 42.5 million tons in 1985–1987 to 63.4 million tons in 1998–2000 (Figure 1b). This increase was due, firstly, to the expansion of the cultivated area (Figure 1a) and, to a lesser extent, to the increase in productivity due to improved agricultural practices, climate-smart agriculture, improved soil quality, use of organic farming practices, and use of precision agriculture, among others. During this period, the average yield increased from 13.7 to 15.8 tons per hectare (15.0%) (Figure 1c).

The production of Musaceae in the LAC region is in the belt that extends from Mexico to Argentina, including Venezuela, Brazil, and the Caribbean islands, which play a fundamental role in its growing production, and throughout history, these crops have prevailed production systems based on two monocultures. The first was during the production of the GM banana, which dominated the world market until 1959 [10], due to the destruction of its plantations by vascular wilt disease caused by the fungus *Foc* Race 1. The second began with Cavendish clones that replaced the previous clone and remain in force until today [25].

From the first report of *Foc* TR4 (the year 1990) and until the first report of its presence in latitudes outside the region where it originated (Asian region) in 2013, 46 years passed, and from that moment until its last report in 2018, five years passed [10]. In August 2019, the ICA confirmed the presence of TR4 in an area of 175 ha in Colombia’s La Guajira department. Symptomatic plants were spotted earlier, and the area was quarantined on 11 June. This was the first report on *Foc* TR4 in Latin America [11]. In 2021, TR4 was reported to be present in northern Peru. The fungus was detected on a 0.5 ha area in the department of Piura, known for its production of organic bananas [23].

It is estimated that *Foc* TR4 has destroyed more than 200,000 ha of bananas in South Asia, and according to recent estimates in the Philippines, more than 15,000 ha were lost in recent years, of which about 11,000 ha belonged to small farmers, which highlights the vulnerability of this sector [32]. As a quarantine pathogen for many countries, the arrival of *Foc* TR4 and the lack of solutions to deal with it, can in many cases lead to the closure of farms and, consequently, to unemployment [33].

Currently, *Foc* TR4 has spread to different continents, causing multi-million-dollar losses, and its recent reports in Colombia and Peru (years 2019 and 2021, respectively) imply that all the countries of the LAC region, with emphasis on Ecuador, Costa Rica, Panama, and Venezuela, are in a state of maximum risk due to the constant flow of people (includes tourists) and commercial transactions, with special attention to those from the Asian continent. A review of the risk analyses in the new scenario is imperatively necessary [10].

### 3.2. A Bibliometric Analysis of the Scientific Production of Foc TR4 in LAC

A total of 284 publications on the topic of *Foc* TR4 were identified in the WOS database between 2018 and 2022, including 243 (85.6%) original research articles, 11 (3.9%) review articles, 22 (7.7%) conference papers, 8 contributions from other forms of publications, including notes, book chapters, and erratum. Of all the published articles, only those that mentioned the LAC problem or according to the origin of the authors in the WOS central database were selected, obtaining a total of 92 documents, with contributions from Brazil (22.8%), Colombia (13.0%), and Mexico (7.6%), which were the highest proportions, followed by the scientific production of Costa Rica, Cuba, Ecuador, Peru, Uruguay, and Venezuela.

The bibliometric analysis of the keywords is shown in Figure 2a. Keywords provided by the article authors and occurring more than 5 times in the WOS core database were enrolled in the final analysis. Of the 1074 keywords, 37 reached the threshold. The keywords that appeared the most were “Fusarium” (total bond strength 119) and “Musa” (total bond strength 103), which had a strong link with “microbiology” and “plant disease” (Figure 2a). Four keyword clusters were identified. The first cluster with 7 items had the keyword “non-human”, which in this context, is an expression used by paper authors to describe any organism that does not directly affect human health (link 19 and total strength of link 78) and was the one with the greatest influence and occurrence (17 times), followed by “Fusarium wilt” (link 20, total bond strength 59, and occurrence 33 times). The second cluster had 7 items and was “plant disease” (link 19, total strength of link 81, and occurrence 14 times); “Musa” (link 18, total link strength 103, and occurrence 21 times) and “microbiology” (link 18, total link strength 80, and occurrence 12 times) had the greatest influence on the network. The third cluster had 5 items, with “fungi” being the most influential term (link 17, total link strength 57, and occurrence 22 times). Finally, the fourth cluster had only 4 items, with “pathogenicity” being the most influential term (link 16, total strength of link 47, and occurrence 8 times) followed by “cultivar” (link 16, total strength of link 46, and occurrence 7 times).

The bibliometric analysis of topics and trending topics is indicated in Figure 2b; three subjects of *Foc* TR4 studies were found. Group 1 involves publications describing the diagnosis of *Foc* TR4 and the characteristics of the cultivar. Group 2 involves publications in the genetic field and pathogenesis. Group 3 involves contributions that investigated the effects on growth and control. Figure 2b shows the network map of trending topics based on keywords used from January 2018 to September 2022. The indicator shows current posts from purple to yellow. More studies focusing on the effects on growth and biological control, for example, have recently been published.

The threat of *Foc* TR4 remains latent and at risk of spreading throughout LAC. Until now, the new cases of *Foc* TR4 reported in Colombia and Peru made the situation very worrying. To fight this disease, academia in LAC with international allies must join this “battlefield” as soon as possible to provide recommendations and suggestions to prevent the appearance of the disease in Musaceae.

The contributions in LAC demonstrate a proactive research community in search of the effective control of *Foc* TR4. Together, the documents provide an overview of the current understanding of the biology and pathogenesis of TR4, its management, and control alternatives; in this sense, integrated and innovative solutions are required to be adopted by all interested parties to build the sustainability of Musaceae systems in the future. There are some topics in which further development is expected, such as the use of new resistant genotypes [35]; the identification of new virulent strains [10]; the environmental suitability of the disease [13]; and the diversification of the genetic bases of the banana industry combined with other types of solutions, such as soil management or the integration of bananas with other crops [36].

Despite the advances in the knowledge of *Foc* TR4 and its behavior in Musaceae, thanks to the excellent work and efforts of world-renowned researchers, there is still much to learn about this soil fungus. Llauger et al. [27] recommended the intensification of research on *Foc* TR4 in LAC, especially in aspects related to hosts [37], epidemiology [13,35], and control [38]. However, the unexpected recent detection in Colombia and Peru, with different situations in each of these countries, means that specific resources must be dedicated throughout the region to investigate the risks and specific problems posed by *Foc* TR4 in each country and crop and about the best methods to address them. The bibliometric analysis of the scientific production of *Foc* TR4 in LAC shows that there is a growing interest in this topic. The number of publications on TR4 has increased in recent years, indicating that researchers in the region are actively working on finding solutions to this problem with innovative techniques based on artificial intelligence [39,40], aerial images [18,41], and machine learning methods [42,43,44].

The experience of the problems that have occurred in LAC, with the opposition to disease control measures based on eradication and containment by certain sectors, makes this research especially necessary in the case of this highly mediated fungus. Educational institutions, scientific societies, and public research organizations still have a long way to go to encourage science to permeate society, and political decisions to address serious problems such as *Foc* TR4, which threatens the common welfare, are based on knowledge and not only on opinions or reflections.

### 3.3. The Current Situation of Foc TR4 in Colombia, Peru, and Venezuela

#### 3.3.1. Current Situation in Colombia

In Colombia, the location of the confirmed outbreaks is shown in Figure 3, according to official data from the ICA [45]. Currently, in farms with the confirmed presence of Foc TR4, quarantine measures ordered by the ICA are maintained, such as restriction of movement of people, the restriction of movement and use of farm equipment, vehicles, and containers, and the use of products for disinfection at different control points in the quarantine area. A calculation of the cost of implementing quarantine and biosecurity for the first year in farms with organic production, in the department of La Guajira, was estimated at USD 350 per ha/year, with USD 160 ha/year being the cost of its maintenance [46].

Figure 4 shows the evolution of the registry of farms under quarantine of ICA due to confirmed positive outbreaks for *Foc* TR4 between June 2019 and December 2022. The current area under quarantine corresponds to 2282 ha in the department of La Guajira and 854 ha in the department of Magdalena, Colombia; the total is 3136 ha [45].

#### 3.3.2. Program for Prevention and Containment of Foc TR4 in Colombia

In August 2019, the NPPO of Colombia reported their first detection of *Foc* TR4 in the La Guajira department. Consequently, the NPPO activated a prevention and containment program called “From theory to action”, which consists of six core actions and counts on solid governance to respond to the threat that Foc TR4 represents in Colombia. In preparation for the response, the NPPO reinforced the program based on experiences addressed during a simulation exercise conducted by FAO and the International Regional Organization for Agricultural Health (OIRSA) in Costa Rica in May 2019. The core activities involve:(a)Program supervision: A Unified Command Post (UCP) was created by instruction of the President of the Republic and the Minister of Agriculture. The national agriculture authorities, banana farmers’ associations, the military, and police forces formed the command to coordinate the interinstitutional actions for the containment of *Foc* TR4 in La Guajira, Colombia (Figure 5).(b)Articulation and coordination with stakeholders: The program involved the action of the Colombian Farmers’ Association (SAC), banana local farmers’ associations, and other actors in the supply production and commercial and logistics chains.(c)Empowerment and mandate: The NPPO was empowered to lead the response, take quarantine measures to contain the outbreak, be accountable for the program execution, and act as the technical secretariat of the UCP.(d)Take advantage of national capacities: The capabilities of the NPPO were used, using the national research institution of Colombia (AGROSAVIA), Bioversity International and the International Center for Tropical Agriculture (CIAT), Cenibanano, academic institutions, and other institutions dedicated to agricultural technology transfer. These institutions proposed sustainable alternatives based on scientific evidence and appropriate risk communication to respond to the *Foc* TR4 presence.(e)Cooperation and technical assistance: recognized organizations and world experts in *Foc* TR4 provided sustained support, cooperation, and technical assistance.

**Figure 5 pathogens-12-00277-f005:**
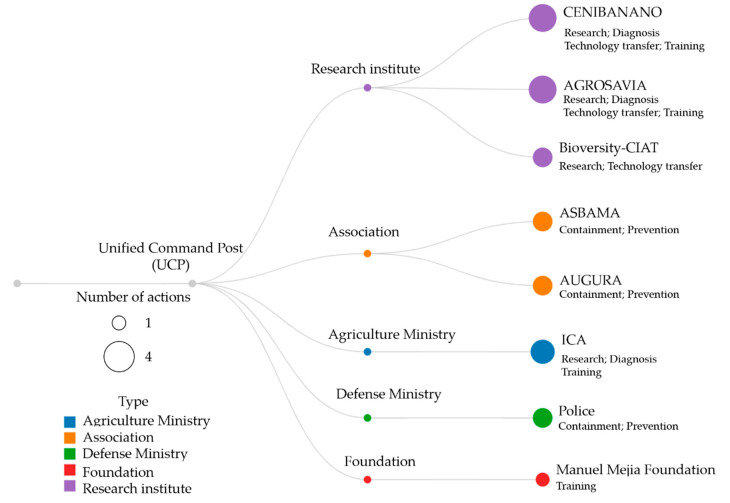
Type and number of actions carried out by the entities that make up the Unified Command Post (UCP) in Colombia.

In Colombia, the public–private agreement for the implementation of the above actions and to keep the Unified Command Post UCP informed remains in force. One of the lessons that left the attention of the first outbreaks reported is the solidarity of institutions and experts from around the world. From countries such as El Salvador, Panama, Cuba, Ecuador, Brazil, Italy, and Australia, the assistance provided has been of high value.

In 2021, Colombia published updated protocols for importing germplasm and banana propagation material [47] and consolidated new analytical and plant diagnostic capabilities in the regions of Urabá (Antioquia) and on the North Coast of the country, with a new laboratory for the detection of *Foc* TR4, built with joint contributions from the government and producers [47]. Augura, Asbama, ICA, Bioversity-CIAT, and other institutions included the use of drones with multispectral cameras for phytosanitary surveillance, the characterization of risk areas, and the search for better dispersion models of *Foc* TR4 [48]. The objective is to have accurate and real-time information on large areas of export farms and local production areas.

The strategy “From Theory to Action” made it possible to generate information for the construction of the Regional Strategy and Action Plan for the Preparation, Prevention, Detection, Response, and Recovery of Latin America and the Caribbean to Fusarium Wilt of the Musaceae—Tropical Race 4”, published by FAO [27]. Colombia continues to leverage national capacities and international assistance to strengthen its Foc TR4 containment strategy.

#### 3.3.3. Current Situation in Peru

In December 2019, the National Agricultural Health Service of Peru (SENASA) declared a phytosanitary alert throughout the national territory for the pest *Foc* TR4 [49], this official declaration was given because of the confirmation of the first detection of *Foc* TR4 in Colombia by the NPPO of Colombia. In April 2021, SENASA declared a phytosanitary emergency throughout the national territory due to the presence of the pest *Foc* TR4 in the Piura region [50]. On 10 April 2021, the Plant Health Directorate reported the presence of the pest *Foc* TR4 in the Chocan sector, Querecotillo district, Sullana province, and Piura region, a situation that determined the need to execute phytosanitary measures to avoid the spread of the pest to other areas of the national territory [50]. 

Figure 6 shows the evolution of the registry of plants in quarantine of the National Agricultural Health Service of Peru (SENASA) for 94 confirmed positive outbreaks of Foc TR4 between April 2021 and June 2022. The intervened area corresponds to 84.91 ha, and the area currently in quarantine corresponds to 1.44 ha in the province of Sullana, Piura Region.

The molecular phylogenetic analysis of TR4 isolates from Colombia and Peru showed that it is closely related to TR4 isolates from other countries, indicating that the pathogen likely spread to these countries from a common source. Epidemiological evidence suggests that the TR4 race invaded Colombia and Peru through the importation and planting of infected banana planting material. The TR4 fungus can persist in the soil for long periods and can be spread through contaminated soil, water, and farm tools.

#### 3.3.4. Program for Prevention and Containment of *Foc* TR4 in Peru

The NPPO, in the Piura region, promoted the formation of a technical command for the containment of *Foc* TR4, to coordinate, evaluate, and adopt the necessary measures for the containment of the Fusarium Race 4 Tropical disease that attacks bananas, detected in the district of Querecotillo.

An action plan oriented to the phytosanitary surveillance of the *Foc* TR4 was elaborated through the prospection of banana and plantain-producing areas at the national level, and it included the following: the laboratory diagnosis of samples of suspicious vegetative material collected from banana areas; the control of outbreaks that consists of attending in a timely and rapid manner the outbreaks of the plague that could occur; quarantine actions aimed at intercepting vegetative material to prevent the entry and spread of the pest, for which external control posts were strengthened and joint operations were carried out with the participation of the National Police of Peru, Public Ministry, and National Superintendence of Customs and Administration (SUNAT); and the strengthening of capacities and communication to train the actors of the banana chain on the importance of the pest and the biosecurity measures that they must implement in their fields to prevent the fungus from entering other regions of the country.

In 2021, the NPPO, with the support of its technical team, carried out activities in the Piura region to prospect and sample suspicious plants for analysis in the laboratory, prioritizing the banana-producing areas close to the first focus of *Foc* TR4 in the Chocan sector, and similar actions have been implemented in different banana- and plantain-producing regions in the country. It is important to indicate that, in Peru, there is 160,000 ha of Musaceae production.

Initial isolates obtained from samples of banana plants with symptoms confirmed the first incursion of *Foc* TR4 in Peru [23], and the dispersal and confirmation of new plants with *Foc* TR4 developed in organic banana production areas. 

There is 10,217 ha of certified organic bananas whose production is exported to the US market, Europe, and Asia. The province of Sullana represents 65.84% of the total area of organic bananas, and the production fields are concentrated in the Chira River valley. The organic banana chain is characterized by the fact that it is cultivated by producers who own areas between 0.25 and 1.0 ha, grouped in organizations that export the fruit through export companies. 

The organic banana production fields in the Chira River valley are made up of small lots that are not separated by live or dead fences, the irrigation system is superficial by flooding, and the water comes from the Poechos dam and is distributed to the production fields through irrigation channels, and the rainfall per year is not greater than 150 mm. 

Peru is the second country in Latin America that, in less than 16 months, went from a state of alert to a state of phytosanitary emergency due to the confirmed presence of Foc TR4 in the organic banana-producing areas for export located on the north coast. Although Foc TR4 is currently contained in the province of Sullana, the spread of new outbreaks in organic banana production fields is gradually increasing. The last record of high rainfall caused by the El Niño/Southern Oscillation phenomenon occurred in 2017 in Piura, significantly affecting banana plantations with floods. There is a risk that, if a new El Niño phenomenon occurs with high rainfall, the dragging of soil and bushes in the fields with Foc TR4 could accelerate the dispersion and appearance of new foci. Currently, the actions of the NPPO have focused on permanent surveillance, sampling, analysis, and laboratory confirmation if the suspicious plants are positive.

There are different organizations of organic banana producers grouped into associations and cooperatives that carry out harvests through crews of workers who travel to different production areas. Faced with a scenario of family farming in small areas of banana production, initiatives are being developed to develop a biosafety program for small farms that helps contain the spread of *Foc* TR4 within the Piura Region and does not spread to other regions of the country. 

The accompaniment of national and international cooperation has been essential since the confirmation of the presence of the disease, as well as the financing of new projects that help develop a research agenda, technical assistance, and phytosanitary monitoring (Figure 7). The National Institute of Agrarian Innovation of Peru (INIA) initiated procedures for the importation of varieties of Musaceae with resistance and tolerance to *Foc* TR4, and these tests that are carried out under controlled conditions and in the field and require no less than 5 years, which is why a comprehensive strategic plan based on four pillars, biosafety, monitoring, research, and technology transfer, is required, where public and private entities, the producer sector, and the academy develop their functions in an integral and coordinated manner.

The development of an efficient biosecurity model for small farms was recently designed, and its validation and implementation will take a while; meanwhile, the strategy that is promoted by public and private entities is aimed at raising awareness among organic banana producers about the risk of this disease, its different dispersal mechanisms, and the importance of containing it to avoid its spread to other regions where banana and conventional plantain areas are large and are a source of food security. INIA initiated the process of a strategic alliance with international entities for the introduction of resistant varieties; in parallel, they are in the process of designing a mechanism for the distribution of plant material.

#### 3.3.5. Current Situation in Venezuela

In January 2023, the presence of *Foc* TR4 in Venezuela was officially declared for the first time according to the declaration of a state of phytosanitary emergency information provided by INSAI (INSAI official written communication, 19 January 2023). The sources of infection were found in the states of Aragua, Carabobo, and Cojedes, which is indicative of the advance of the pathogen in the American continent. However, no further scientific information has been published to date. In this regard, pertinent measures and reinforced biosafety protocols have been taken to try to contain or delay its progress.

### 3.4. Medium-Term Prospects in LAC Member Countries

The introduction of edible Musaceae to the American continent, at the time, was conceived as a solution to the economy of the plantations of the new world for the feeding of local inhabitants, and they later became indispensable, traditional products and essential elements in the economy of many countries [51]. Its production and export process in LAC began at the end of the 19th century, thanks to the acceptance in the American market of the banana fruit clone GM, which, due to its growing demand and strategic location of this market concerning the LAC region, motivated the investment and anchorage of large North American banana companies in the area, which began in Central America and later spread to different countries of the continent [51].

The action of *Foc* led to the GM banana being extended as a monoculture, causing the plantations to be extremely vulnerable to disease outbreaks, as there is a great diversity of harmful organisms that co-evolve with bananas [52], which happened with the fungus *Foc*. The first report of *Foc* on banana cultivation was in 1874 in Australia, through race 1, and its effects were overcome by a change to Cavendish cultivars (Williams, Gran enano, Valery, Pineo Gigante, among others) resistant to this race, representing more than 90% of banana exports in recent decades [53]. However, this substitution was carried out on the same organizational structure of the planting systems established for the time with the traditional clone (GM), adjusting population densities in order to exceed or failing to maintain the yield and existing production indices, and they also managed as monocultures with their different clonal variants.

At present, hundreds of edible Musaceae genotypes with the potential to be cultivated have been identified. However, of this great diversity, only a limited fraction is consumed globally according to taste and cooking preferences [52,54]. World production is based on a few clones that essentially belong to three genetic groups (AAA, AAB, ABB) and that are slightly differentiated from each other by some clonal variations. Among them, the dominance of Cavendish cultivars produced in global monocultures clearly leaves plantations extremely vulnerable to disease outbreaks due to the existence of a great diversity of harmful organisms that co-evolve with bananas, just as could happen in other crops [52,55].

Currently, the banana sector presents serious concerns regarding the destructive potential of Fusarium in the tropics, which is different and more serious than that previously reported in the subtropics [53]. Similar to all other soil-dwelling strains of *Foc*, TR4 cannot be managed with fungicides or eradicated from the soil with fumigants. Its ability to survive decades in the soil, its lethal impact, and its wide range of hosts make it cataloged as the greatest threat to the production of bananas and plantains produced in tropical and subtropical conditions in the last 20 years [53,56].

It is estimated that *Foc* TR4 has so far affected more than 200,000 ha in countries where it is endemic and where it has been introduced due to the absence of effective sanitary measures for its contamination. Similarly, in recent years, more than 15,000 ha was lost to *Foc* TR4 in the Philippines, of which some 11,000 ha belonged to small farmers, highlighting the vulnerability of this sector [35,57]. The FAO indicates that more than 135 countries worldwide cultivate edible Musaceae in the tropics and subtropics, and most of the producers are farmers who use this product for domestic consumption or local markets because less than 15% of world production is exported [58].

The eventual entry of this pathogen into territories where the pest is absent would generate devastating consequences in terms of food security, economic and social stability, as well as uncertainty and confidence in exports [59]. In this last scenario, the response of the banana sector in the LAC region, promoted by the Latin American and Caribbean Research Network for the Development of Musaceae (MUSALAC), began in the year 2000, with various actions and activities to understand the potential threat of this disease to the American continent and its serious consequences [57].

Subsequently, the OIRSA became involved, and for the first time, this issue was considered and analyzed in depth. The decision was made to contact the phytosanitary authorities of the countries to put this pathogen on the list of quarantine pests and start campaigns for the prevention of *Foc* TR4 in the region. A direct connection was created between the research institutes and the National Phytosanitary Protection Organizations (NPPOs) to build a legal framework based on scientific information (from molecular diagnostic methods to the efficient use of disinfectants for biosafety on farms) that would deal with a plague such as this one [57].

At this converging point (NPPOs and scientists), the FAO, Bioversity International, and Plant Health Research Institute (INISAV) of Cuba became involved, starting the training of officials from official laboratories and research centers for the detection, diagnosis, and collection of samples and management of the disease, in addition to carrying out drills to detect and manage possible outbreaks, among other details, in the different countries [57].

At the same time, a Contingency Plan (PC) was prepared, which was published in 2013, and it constitutes a base document for each country to create its national plans adapted to its scenarios to implement necessary actions and avoid, as much as possible, the greater damage to the Musaceae industry when the fungus arrives on the continent [57]. All of this made it easier for the NPPOs to make decisions and take actions when Foc TR4 outbreaks were detected in Colombia and Peru, achieving a better response capacity from molecular laboratory diagnosis to contain the foci.

The success obtained translates into managing to contain the advance of the pathogen and to limit it to those regions, which was possible due to the previous preparation and due to the level of commitment and integration of those countries. The search and identification of focal points in each country continue to be a high priority, from Argentina to Mexico, including the Caribbean islands [57].

The conclusions of various workshops and products of the FONTAGRO project in Colombia indicate the need for countries to work on the exclusion, diagnosis, evaluation, and development of resistant materials, as well as on the epidemiology and integrated management of the crop that allows saving time while progressing in obtaining technologies that allow for the mitigation of the impact of *Foc* TR4 in the region [60].

The actions and activities carried out in the other banana-producing countries of the LAC region must be oriented and based on references and/or lessons learned from the efforts undertaken in Peru and Colombia about the *Foc* TR4 outbreaks, as an effort to buy time in the face of problems already present in Latin America and the Caribbean. Developing articulation between countries and institutions achieves a greater capacity for building.

In Central America, the NPPOs of the different countries, with the support of national institutions, OIRSA, FAO, and mixed or private institutions such as the National Banana Corporation (CORBANA) of Costa Rica and the Honduras Foundation for Agricultural Research (FHIA), work together to establish preventive measures to avoid the entry of *Foc* TR4 and the management of the use of canine units to reinforce baggage controls in ports and airports and the entry of merchandise stands [60]. Similar situations take place in South America, where the management of Agrosavia in Colombia, SENASA, and INIA in Peru stand out, with the support of the Bioversity International alliance—CIAT, FAO, OIRSA, and The Inter-American Institute for Cooperation on Agriculture (IICA), among others. In Brazil, it is managed through the Ministry of Agriculture supported by Embrapa, The National Institute of Comprehensive Agricultural Health (INSAI) in Venezuela, the National Institute of Agricultural Research of Venezuela (INIA), and the Venezuelan Institute of Scientific Research (IVIC), among others. Fortunately, there have been no additional reports in any other country so far.

Due to the location of *Foc* TR4 outbreaks in LAC, the risk levels (in theory) increase for Ecuador, Venezuela, and Panama. However, due to specific circumstances, two countries, in particular, deserve special attention:(a)Ecuador: It is considered the main banana exporting country worldwide and is among the two countries that have reported the disease. The Phytosanitary and Zoosanitary Regulation and Control Agency—AGROCALIDAD in quality, NPPO of Ecuador, in Resolution No. 122 of the year 2017, designated *Foc* TR4 as a quarantine pest for the country [61].

Because once its presence in Colombia and Peru was confirmed, the biosecurity measures that were implemented since 2011 were maintained and strengthened at official control points (ports, airports, and border crossings) as well as at the barrier blocks and control established by the agency to prevent and avoid their entry to safeguard the production of Musaceae in Ecuador [61].

Similarly, the agency emphatically ratified the general and specific phytosanitary requirements for the importation of in vitro Musaceae plants, in addition to prohibiting the entry of germplasm and any type of reproductive material (Musaceae) from countries where race 4 is reported (Resolution No. 048). They also established, as a mandatory phytosanitary measure, the external disinfection of containers that enter Ecuadorian territory; regardless of the cargo they contain, disinfection is carried out (Resolution No. 145) [61].

During the year 2021, the AGROCALIDAD National Surveillance System at the national level monitored for *Foc* TR4 in farms producing Musaceae 6496 times, and 116 samples of plants with suspicious symptoms were sent to the molecular biology laboratory, all with negative results [61].

(b)Venezuela: The recent detection of *Foc* TR4 in the central and central-western region of Venezuela as well as to the proximity of the first source of infection in the Colombian Guajira to the border of both countries, where there are conditions of high fragility due to the flow of people, the commercial exchange added to the possible overflow of rivers in adjacent areas due to the of rainfall caused by tropical depressions, increasing the level of dispersion risk [10].

The INSAI as the National Phytosanitary Protection Organization (NPPO) of Venezuela, based on its Administrative Ruling 41,480 of the year 2018 about *Foc* TR4, implemented the “Program for the prevention, detection, management, and control of Fusarium wilt in Musaceae caused by *Foc* TR4 for the Bolivarian Republic of Venezuela”. It includes epidemiological surveillance with an emphasis on border areas and the activation of sampling at the national level for processing and analysis through PCR [62]. During 2022, Venezuela was accompanied by the Global Alliance against TR4, supported by IICA, and it was their mission to support the banana sector in facing the challenges of the disease through the development of knowledge [10,13], technologies, and mechanisms that allow for finding a definitive scientific solution that favors the eradication of the fungus [63,64,65].

Different activities and common actions were developed by most LAC countries against *Foc* TR4, where the following stand out:(a)Campaigns on the radio, television, and social networks about the impact that the plague could have, both for the productive sector as well as for the population in general. They try to create a process of awareness of the population, in general, about the risk that the illegal introduction of plants, leaves, “seeds”, or souvenirs of vegetable fibers entails to the country in reference.(b)Placement of technical signage on symptoms and biosafety measures in communal and strategic areas.(c)Educate, inform, and support the implementation of relevant biosecurity measures.(d)The capacities of the official Pest Diagnosis Laboratory were strengthened (in terms of human resources, equipment, and methodology).(e)The inspection of imported in vitro plants that have entered each country in recent years, verifying that they were free of the pest.(f)Research on possible resistant varieties, although there is not yet a commercially viable option.(g)At the airport level, baggage checks and merchandise entry control are reinforced. The purchase of more X-ray equipment for entry points is under management. Special mats are placed with products that deactivate the possible presence of the fungus in the shoes or luggage wheels of people who enter the country through any of the entry points or borders. They are also improving and monitoring the conditions for international waste management (airline waste).(h)Some countries (the largest producers and exporters of bananas) are improving the infrastructure of the fumigation arches at the border and are updating the type of disinfectant used.

In the medium term:(a)Consider that all Musaceae production sites must implement biosecurity measures in a time not exceeding one year.(b)Strengthen the surveillance system in all Musaceae production sites, with emphasis on actions to reinforce phytosanitary inspections in ports, airports, and border crossings.(c)Give continuity to the training for producers and other actors in the agricultural production chain so that they know how to implement biosecurity measures on their farms and that they can recognize the early symptoms caused by the pathogen and how to proceed in the event of a possible case of the disease.(d)Strengthen the diagnostic capacity for *Foc* TR4 and other Musaceae pests.(e)Promote the creation and management of germplasm banks that serve as support for the development of breeding programs and that contribute to the diversity of clones in planting systems and the evaluation of new resistant or tolerant materials.

The medium-term prospects for the banana industry in LAC member countries are uncertain, as the fungus is difficult to control once it is established in a field [36]. Measures such as quarantine, the destruction of infected plants, and the use of clean planting material are implemented by the governments of LAC member countries, but it is hard to control the spread of the fungus. Research on developing resistant varieties [21] and improving soil management practices is ongoing [22,66,67], but it is uncertain if this will be enough to mitigate the impact of TR4 on the banana industry in the region.

### 3.5. Impacts of Foc TR4 on World Trade and Local Food Security in LAC

#### 3.5.1. Impact of *Foc* TR4 in Countries with the Disease

The introduction of *Foc* TR4 in a given country does not necessarily lead to the total disappearance of production or of the export sector. Indeed, in the case of the Philippines, where the disease was first detected in 2005, it was reported that around 15,700 ha of banana plantations were affected out of a total of 440,000 ha [68]. This means that the country has not only managed to avoid a massive spread of the disease but also to remain among the top world exporters of bananas. In 2021, more than 15 years after the initial detection of the fungus, the country ranked number three in world banana exports (Table 1).

Although the Philippines example offers an optimistic outlook, in other regions, such as in China’s Guangdong and Hainan provinces, the disease is reported to have spread to 70 percent of plantations [70]. Annual economic losses caused by TR4 were estimated at USD 121 million in Indonesia, USD 253 million in Taiwan, and USD 14 million in Malaysia. On the infected farm in Mozambique, TR4 caused severe damage to the 1500 ha plantation within four years of the first detection of the disease, and the farm was forced to cease operations [71,72]. Thus, the range of impact of the disease is large and proportional to the disease’s contention and the producer’s financial capacity to face simultaneous yield losses and increased production costs. 

#### 3.5.2. Potential Short-Term Impact on World Markets

In Latin America, TR4 has been so far reported in the Guajira and Magdalena regions of Colombia [11] and the Piura region of Peru [23]. Coincidentally, both regions are specialized in the production of organic bananas for export [73,74,75]. Thus, one of the first affected markets would be that of the organic banana. Colombia and Peru are major players in the world organic banana trade (Table 2). Of the nearly 60,000 ha of organic bananas for export reported worldwide, Peru’s organic production occupies nearly 10,217 ha, while Colombia’s is 4000 ha, ranking, respectively, in third and fifth positions among world producers of organic bananas [73].

However, since the detection of TR4 in Colombia in 2019, no significant changes in the dynamics of organic export from this country have been reported to date. On the contrary, exports of organic bananas from Colombia seem to have continued to develop because of the sector’s growth [76] and the effective containment of the disease’s spread. In the case of Peru, the detection of *Foc* TR4 in 2021 took place in the context of the crisis in the sector. Organic exports from this country were decreasing since 2018 because of rising competition in export markets and economic difficulties faced by small producers and the exporter sector. It could be expected that the arrival of the TR4 aggravated this waning dynamic. 

As it has been reported in other countries, once a farm has been contaminated, managing the disease is extremely challenging and costly for producers. TR4 leads to simultaneous yield losses and increased production costs. Moreover, the potential repercussions of infection by TR4 are of even greater concern to organic banana production, since organic agricultural practices do not permit genetic modifications, leaving classical breeding of disease-resistant cultivars as the only option for adaptation [71]. In this context, if the disease were to further spread in the production areas of Peru and Colombia, a progressive decrease in exports of organic bananas could be expected in the mid-term because of lower yields, competitivity losses in export markets, or even total ceasing of operations in particular for smallholder banana producers, who often lack the financial means to sustain operations. Although both countries are major players in the world organic banana trade, lower exports from both countries might not lead to an aggravated worldwide deficit of organic bananas since other countries are quickly expanding their organic production sectors and could compensate for the decreasing quantities. This is the case, for instance, in countries such as Ecuador. 

#### 3.5.3. Potential Medium-Term Impact in Peru and Colombia

In the case of non-contention of the disease and a wide spread of TR4 to other banana-producing areas in Colombia and Peru, conventional bananas for export and local food security could be at stake. Exporting nearly 2 million tons of bananas in 2021, Colombia is the fifth world exporter of conventional bananas (Figure 8) [77].

More than 80,000 ha in the country were planted in 2019 with dessert bananas, with 60% of the country’s production destined to export markets and 40% for local consumption [78]. At the same time, over 500,000 ha of plantains were also planted in the country, and they were destined for both local and export markets [79]. 

Most producers in the country have, on average, 5.0 ha of bananas and 3.5 ha of plantains, making them mostly smallholders. Thus, a potential decrease in production because of the spread of TR4 could have a significant impact on both conventional banana export markets and, above all, on local food security. In the case of Peru, since the country does not export conventional bananas, the spread of TR4 to other banana-producing regions of the country implies a major threat to local food security, as it is the main outlet for production. 

The impact of TR4 on the banana industry in LAC member countries could have a significant impact on export trade and local food security [7,28,79]. Many LAC member countries are heavily dependent on banana exports for their economies, and a decline in productivity and profitability in the banana industry could hurt the overall economy of the country [80,81]. Additionally, a decline in banana production could also lead to a decline in local food security, as bananas are a staple food for many people in the region.

## 4. Summary Points and Future Issues

(a)The containment measures for *Foc* TR4 worked until the moment they were defeated by the indiscriminate transit of contaminated material (vegetable soil, water, among others), to which must be added the great expansion of the area planted with Cavendish clones in monoculture systems, highly vulnerable to this pathogen. Its recent foray into LAC presupposes an impact on world banana production and trade.(b)The *Foc* TR4 prevention and containment programs that are currently available must be complied with, but at the same time, it is necessary, as previously mentioned, to advance the knowledge of the multiple factors related to this situation since, in this way, it will be possible to design management strategies for this disease based on scientific knowledge.(c)In Colombia, rapid intervention based on the eradication of areas adjacent to affected lots has been a successful measure. In Peru, the eradication measures have been applied in a localized way to try not to affect the total production of the producers since their production units are very small and represent their main economic support. In Venezuela, the recent emergency declaration of *Foc* TR4 puts local banana production at risk. In these cases, the design of an efficient biosecurity model for the production units, access to resistant plant material, soil health, early detection, and phytosanitary education at different levels about this disease are essential to contain it.(d)The range of impact of the disease is large and proportional to the disease’s contention and the producer’s financial capacity to face simultaneous yield losses and increased production costs. In the case of Peru and Colombia, the further spread of *Foc* TR4 could lead to production losses of organic bananas for export. The current situation in the American continent is even more complicated given the recent declaration (19 January 2023) of the presence of *Foc* TR4 in Venezuela. If the disease was to spread further to other regions, conventional bananas destined for both export and domestic markets could be at stake, leading to significant food safety concerns.

## Figures and Tables

**Figure 1 pathogens-12-00277-f001:**
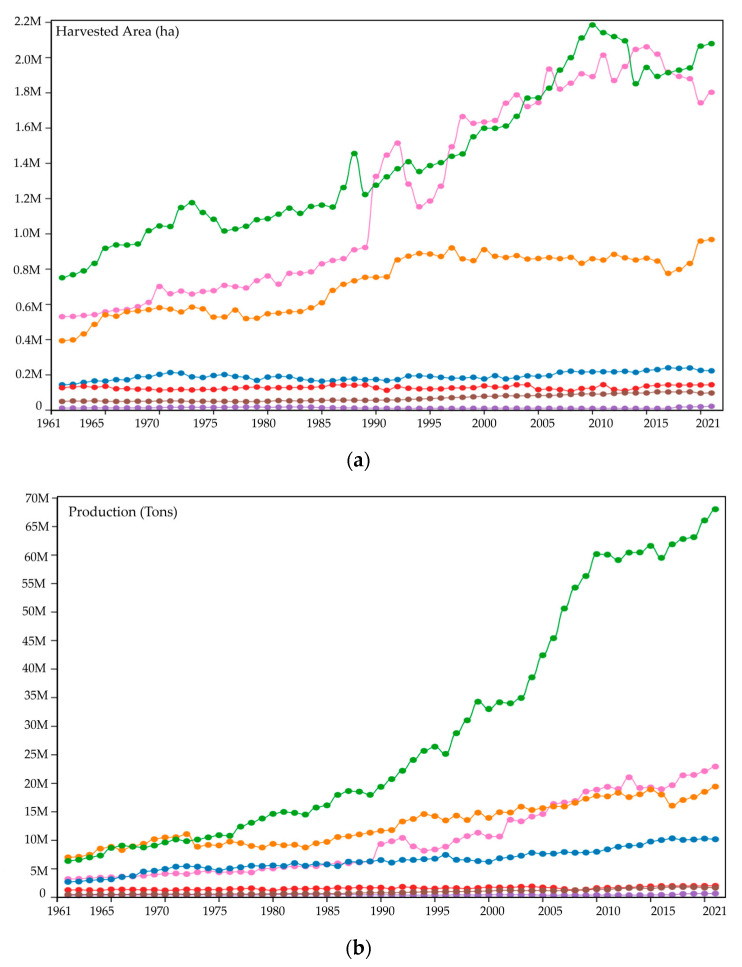
Bump chart of changes in the range of value in (**a**) harvested area (ha), (**b**) production (t), and (**c**) yield (t/ha) during 1961–2021 in Central America, South America, Asia, the Caribbean, Europe, Oceania, and Africa [2]. Note: When interpreting a bump chart, when a line crosses another line, that is indicative of a change in rank. In other words, a crisscross in a bump chart indicates that one entity (continent) has surpassed others in absolute terms, even when the comparison is based on relative ranks.

**Figure 2 pathogens-12-00277-f002:**
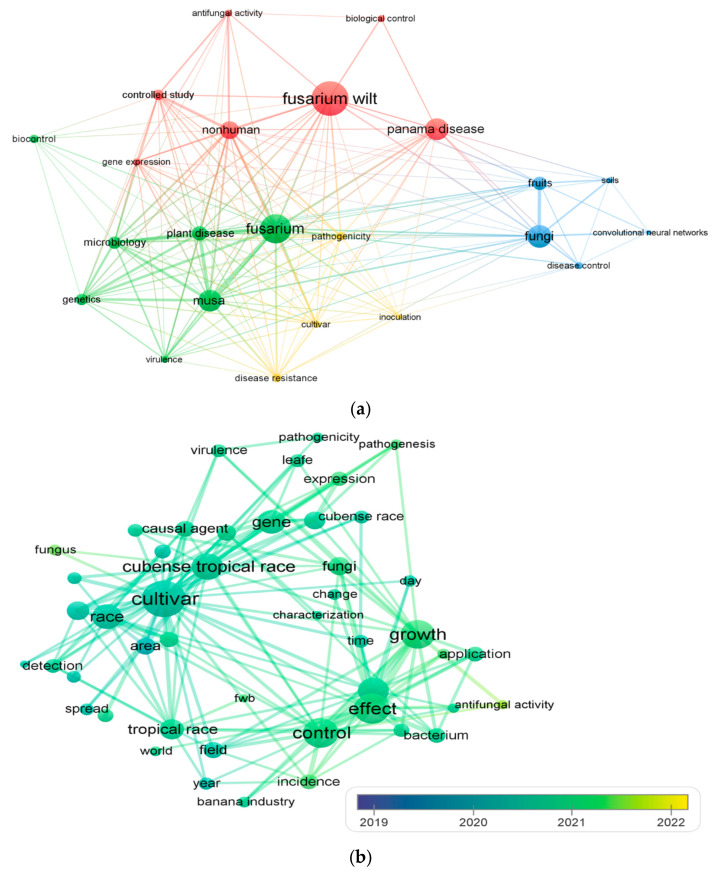
Bibliometric analysis of keywords in *Foc* TR4 publications in LAC. (**a**) Co-occurrence of keywords. The size of the nodes indicates the frequency of occurrence. The lines between the nodes represent their co-occurrence in the same publication. The smaller the distance between two nodes, the higher the number of co-occurrences of the two keywords. (**b**) Network map of trending topics according to the keywords used from January 2018 to September 2022. The size of the circles represents the frequency of occurrence of the keywords. The distance between the two circles indicates their correlation. VOSviewer uses the smart local moving algorithm introduced by Waltman and Van Eck [34].

**Figure 3 pathogens-12-00277-f003:**
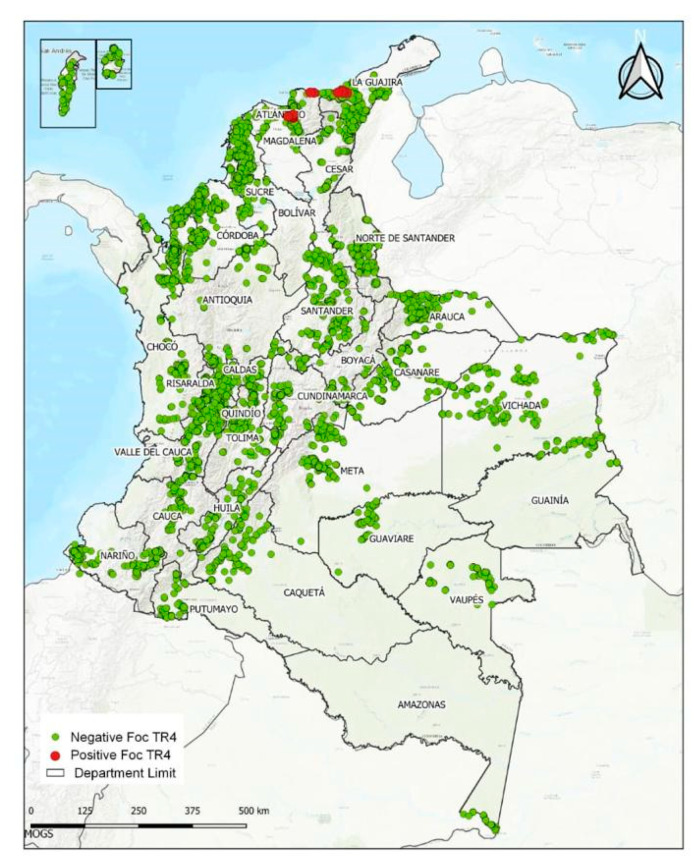
Location of confirmed outbreaks of Foc TR4 in Colombia using PCR protocols and pathogenicity tests according to official data [45]. The collection period was from 1 June 2019 to 31 December 2022, updated in January 2023.

**Figure 4 pathogens-12-00277-f004:**
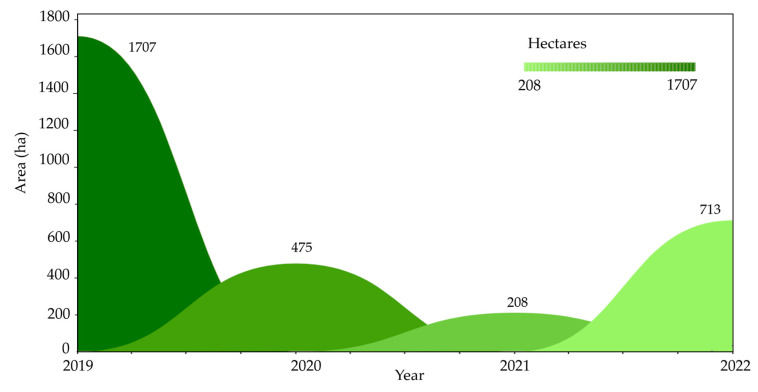
Farms were quarantined due to reports of *Foc* TR4 outbreaks [45] between June 2019 and December 2022 (the last outbreak).

**Figure 6 pathogens-12-00277-f006:**
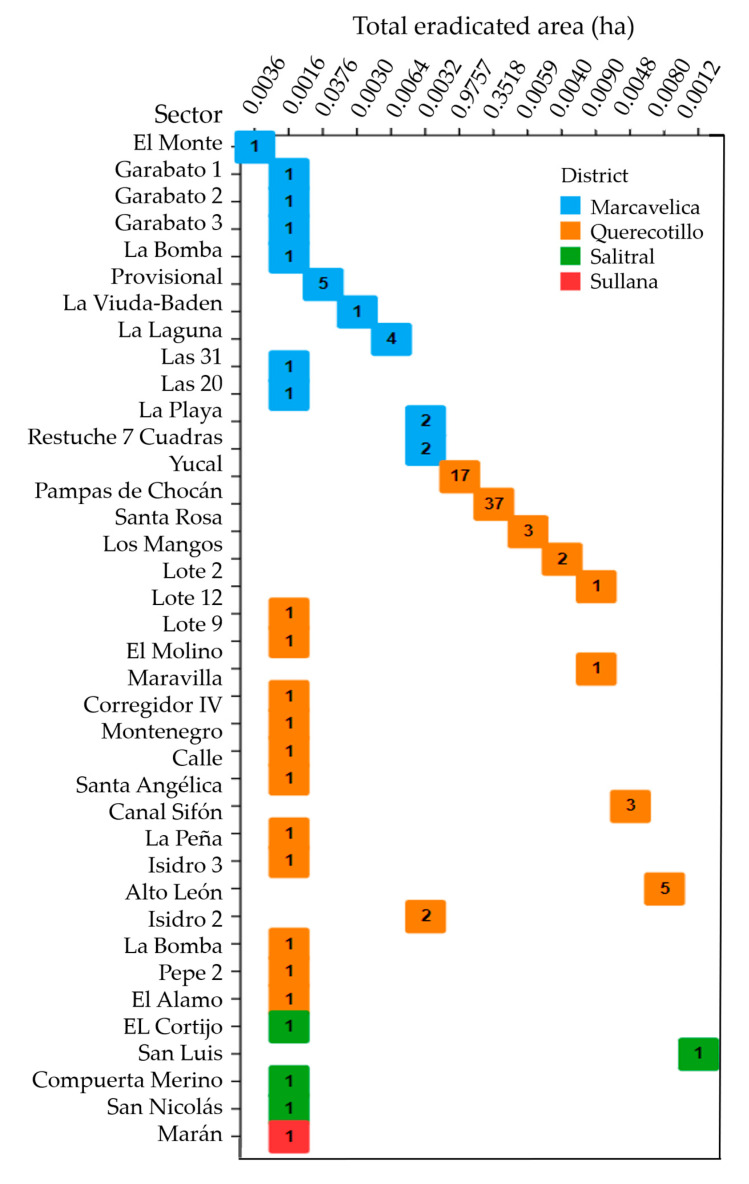
Matrix plot. Comparison of the total eradicated area (ha) by affected sector arranged on the horizontal and vertical axes. Each glyph (square) represents the number of outbreaks of Foc TR4 in the sectors. The color was used to distinguish the districts of the province of Sullana, Piura Region, and Peru.

**Figure 7 pathogens-12-00277-f007:**
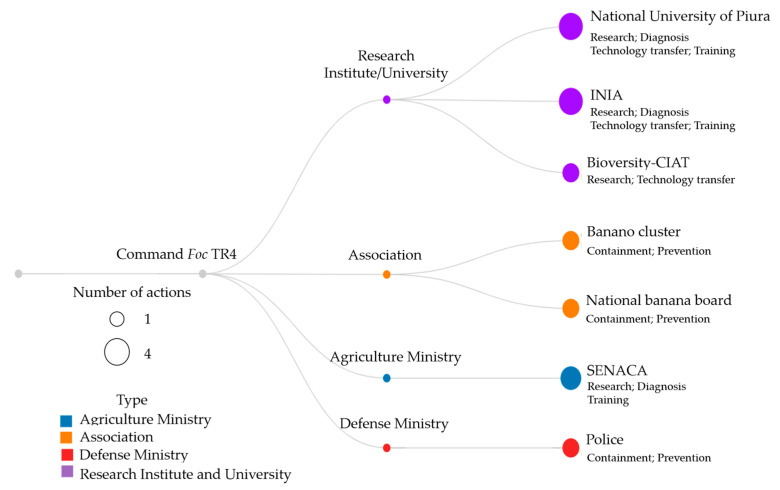
Type and number of actions carried out by the entities in Peru.

**Figure 8 pathogens-12-00277-f008:**
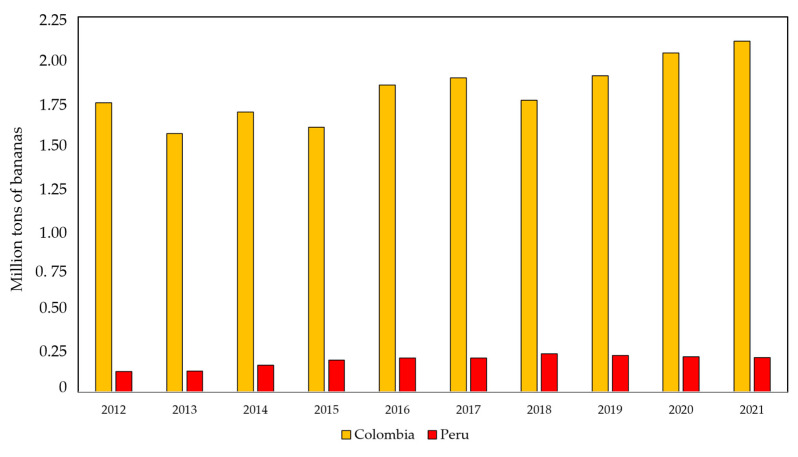
Total exports of bananas from Colombia and Peru to all destinations 2012–2021 [77].

**Table 1 pathogens-12-00277-t001:** Top ten world bananas exported by country in tons according to Fruitrop data [69].

Ranking	Country	Exported Tons in 2021
1	Ecuador	6,915,429
2	Costa Rica	2,507,485
3	Philippines	2,428,887
4	Guatemala	2,290,266
5	Colombia	1,993,597
6	Panama	700,000
7	Mexico	530,481
8	Côte d’Ivoire	380,000
9	Dominican Republic	360,192
10	Honduras	335,968

**Table 2 pathogens-12-00277-t002:** Estimated world organic banana-certified hectares for export according to Fruitrop data [73].

LAC Country	Estimated Area of Organic Banana (ha)	Share of World Surfaces (%)
Dominican Republic	20,000	34.2
Ecuador	18,830	32.2
Peru	10,217	17.4
Mexico	4184	7.1
Colombia	4000	6.8
Other countries (not LAC)	1300	2.2
World total	58,531	100.0

## Data Availability

Not applicable.

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
