# Peer review of "The Advance of Fusarium Wilt Tropical Race 4 in Musaceae of Latin America and the Caribbean: Current Situation"

_pathogens, 2023, doi:10.3390/pathogens12020277_

Round 1

Reviewer 1 Report

Reviewer’s Comments:

The manuscript “Assessing threats posed by Fusarium wilt Tropical Race 4 in Musaceae of Latin America and the Caribbean” is a very interesting work. This paper reports a fungus Fusarium oxysporum f. sp. cubense tropical race 4 (syn. Fusarium odoratissimum) (Foc TR4), causes vascular wilt in Musaceae plants and is considered the most lethal for these crops. In Latin America and the Caribbean (LAC), it was reported for the first time in Colombia (2019) and later in Peru (2021). The objective of this work is to analyze the evolution of Foc TR4 in Musaceae in LAC between 2018-2022. This perspective contains a selection of topics related to Foc TR4 in LAC, which address and describe (i) the threat of Foc TR4 in LAC, (ii) a bibliometric analysis of the scientific production on Foc TR4 in LAC, (iii) the first reports of Foc TR4 in Colombia and Peru, (iv) medium-term prospects in LAC member countries, and (v) export trade and local food security. The results are consistent with the data and figures presented in the manuscript. While I believe this topic is of great interest to our readers, I think it needs major revision before it is ready for publication. So, I recommend this manuscript for publication with major revisions.

1. In this manuscript, the authors did not explain the importance of the importance of the Fusarium in the introduction part. The authors should explain the importance of importance of the Fusarium.

2) Title: The title of the manuscript is not impressive. It should be modified or rewritten it.

3) Correct the following statement “Quarantine and containment were not enough to prevent international dissemination, so it is necessary to find productive varieties to replace Cavendish bananas, one of the strategies to improve the resilience of the export-oriented banana industry, further research should be carried out. on this pathogen and its interaction with Musaceae”.

4) Keywords: The Fusarium is missing in the keywords. So, modify the keywords.

5) Introduction part is not impressive. The references cited are very old. So, Improve it with some latest literature like 10.3389/fchem.2022.1023316, 10.1155/2022/7689617

6) The authors should explain the following statement with recent references, “Augura, Asbama, ICA, Bioversity-CIAT, and other institutions have included the use of drones with multispectral cameras for phytosanitary surveillance, the characterization of risk areas, and the search for better dispersion models of Foc TR4”.

7) Add space between magnitude and unit. For example, in synthesis “21.96g” should be 21.96 g. Make the corrections throughout the manuscript regarding values and units.

8) The author should provide reason about this statement “However, this substitution was carried out on the same organizational structure of the planting systems established for the time with the traditional clone (GM), adjusting population densities, but also managed as monocultures with their different clonal variants”.

9. Comparison of the present results with other similar findings in the literature should be discussed in more detail. This is necessary in order to place this work together with other work in the field and to give more credibility to the present results.

10) Conclusion part is very long. Make it brief and improve by adding the results of your studies.

11) There are many grammatic mistakes. Improve the English grammar of the manuscript.

Author Response

Reviewer 1

The manuscript “Assessing threats posed by Fusarium wilt Tropical Race 4 in Musaceae of Latin America and the Caribbean” is a very interesting work. This paper reports a fungus Fusarium oxysporum f. sp. cubense tropical race 4 (syn. Fusarium odoratissimum) (Foc TR4), causes vascular wilt in Musaceae plants and is considered the most lethal for these crops. In Latin America and the Caribbean (LAC), it was reported for the first time in Colombia (2019) and later in Peru (2021). The objective of this work is to analyze the evolution of Foc TR4 in Musaceae in LAC between 2018-2022. This perspective contains a selection of topics related to Foc TR4 in LAC, which address and describe (i) the threat of Foc TR4 in LAC, (ii) a bibliometric analysis of the scientific production on Foc TR4 in LAC, (iii) the first reports of Foc TR4 in Colombia and Peru, (iv) medium-term prospects in LAC member countries, and (v) export trade and local food security. The results are consistent with the data and figures presented in the manuscript. While I believe this topic is of great interest to our readers, I think it needs major revision before it is ready for publication. So, I recommend this manuscript for publication with major revisions.

  1. In this manuscript, the authors did not explain the importance of the Fusarium in the introduction part. The authors should explain the importance of the Fusarium.

Author: The importance of Fusarium was added in the following paragraph:

Line 82-93: The impact of TR4 on the banana industry in LAC is significant (10, 11, 12, 18). The fungus has caused significant losses in productivity and profitability for banana grow-ers in the region, which could hurt the livelihoods of small-scale farmers and the over-all economy of the countries where banana is a major crop [19]. TR4 is considered a major threat to global banana production, as the fungus can spread easily through contaminated soil, water, and farm tools. The fungus can quickly spread to neighbor-ing fields once it is established in a field, making it difficult to control [12, 13, 20].

To address the threat of TR4, measures such as quarantine, destruction of infected plants, and the use of clean planting material are being implemented by the govern-ments of Latin American countries. Additionally, research to develop resistant varie-ties [21] and improve soil management practices is also ongoing [22].

2) Title: The title of the manuscript is not impressive. It should be modified or rewritten it.

Author: the title was changed: Analysis of the evolution of Fusarium wilt Tropical Race 4 in Musaceae of Latin America and the Caribbean

3) Correct the following statement “Quarantine and containment were not enough to prevent international dissemination, so it is necessary to find productive varieties to replace Cavendish bananas, one of the strategies to improve the resilience of the export-oriented banana industry, further research should be carried out. on this pathogen and its interaction with Musaceae”.

Authors: it was corrected: In conclusion, TR4 is a major threat to banana production in Latin America and the world, and it is important to take measures to control the spread of the fungus and minimize its impact on the banana industry. It's important to keep working on the control of Foc TR4 which requires the participation of the local and international industry, researchers, and consumers, among others, to prevent the disappearance of bananas.

4) Keywords: The Fusarium is missing in the keywords. So, modify the keywords.

Authors: It was added: Fusarium odoratissimum

5) Introduction part is not impressive. The references cited are very old. So, Improve it with some latest literature like 10.3389/fchem.2022.1023316, 10.1155/2022/7689617

Author: the bibliography was updated:

Zheng, Y.; Guo, P.; Deng, H.; Lin, Y.; Huang, G.; Wu, J.; Lu, S.; Yang, S.; Zhou, J.; Zheng, W.; Wang, Z.; Yun, Y. Small GTPase FoSec4-Mediated Protein Secretion Is Important for Polarized Growth, Reproduction and Pathogenicity in the Banana Fusarium Wilt Fungus Fusarium odoratissimumJ. Fungi 20228, 880. https://doi.org/10.3390/jof8080880

Ujat, A.H.; Vadamalai, G.; Hattori, Y.; Nakashima, C.; Wong, C.K.F.; Zulperi, D. Current Classification and Diversity of Fusarium Species Complex, the Causal Pathogen of Fusarium Wilt Disease of Banana in Malaysia. Agronomy 202111, 1955. https://doi.org/10.3390/agronomy11101955

van Westerhoven, A.C.; Meijer, H.J.; Seidl, M.F.; Kema, G.H. Uncontained spread of Fusarium wilt of banana threatens African food security. PLoS pathogens 2022, 18(9), p.e1010769. https://doi.org/10.1371/journal.ppat.1010769

6) The authors should explain the following statement with recent references, “Augura, Asbama, ICA, Bioversity-CIAT, and other institutions have included the use of drones with multispectral cameras for phytosanitary surveillance, the characterization of risk areas, and the search for better dispersion models of Foc TR4”.

Author: [48] Cardenas, J. De la teoría a la acción. Historia del Fusarium R4T en Colombia. Manizales. Capital Graphic SAS, 2022; pp.89.

7) Add space between magnitude and unit. For example, in synthesis “21.96g” should be 21.96 g. Make the corrections throughout the manuscript regarding values and units.

Author: corrected

8) The author should provide reason about this statement “However, this substitution was carried out on the same organizational structure of the planting systems established for the time with the traditional clone (GM), adjusting population densities, but also managed as monocultures with their different clonal variants”.

Author: LINE 621: However, this substitution was carried out on the same organizational structure of the planting systems established for the time with the traditional clone (GM), adjusting population densities, in order to exceed or fail that maintain the yield and production indices existing, also managed as monocultures with their different clonal variants.

At present, hundreds of edible Musaceae genotypes with the potential to be cultivated have been identified. However, of this great diversity, only a limited fraction is consumed globally according to taste and cooking preferences [54, 55]. Whereas, world production is based on a few clones that essentially belong to three genetic groups (AAA, AAB, ABB) and are slightly differentiated from each other by soma clonal variations. Among them, the dominance of Cavendish cultivars produced in global monocultures clearly leaves plantations extremely vulnerable to disease outbreaks, due to the existence of a great diversity of harmful organisms that co-evolve with bananas, just as could happen in other crops [55, 56].

  1. Comparison of the present results with other similar findings in the literature should be discussed in more detail. This is necessary in order to place this work together with other work in the field and to give more credibility to the present results.

Author: corrected. In each section, discussion paragraphs with scientific literature were added.

10) Conclusion part is very long. Make it brief and improve by adding the results of your studies.

Author: Thanks for the observation. However, it is not a conclusion section, it is rather a summary section and final reflection which were summarized according to your observation.

11) There are many grammatic mistakes. Improve the English grammar of the manuscript.

Author: English grammar improved

Reviewer 2 Report

Authors have submitted a review on Assessing threats posed by Fusarium wilt Tropical Race 4 in 2 Musaceae of Latin America and the Caribbean. The paper provides some interesting information. However, a similar paper was found in "Raising awareness of the threat of Fusarium wilt tropical race 4 in Latin America and the Caribbean" published in 2011.

Since it is after 10 years’ time review the current paper may have some interesting aspects. However, it is not highlighted enough though both papers mainly focus on bananas. Authors may mention somewhere since the last compressive review was done in the year, xxxx this paper builds upon the previous findings or anything like that..

In addition, it should be highlighted how difficult it is to identify the pathogen. There are nonpathogenic Fusarium oxys. species.  So any development in molecular identification of the pathogen or are there any morphological key features for race 4? Need to discuss in detail. Starting from morphology to multilocus analysis on species identification.  A phylogenetic tree based on sequence data showing different Fusarium strains found from different countries will give a clear picture of the classification and evolution of the Fusarium race found in LAC regions. 

At the same time, control strategies and pathogen defense mechanisms should be highlighted. 

 Can Table 1 be in graphical format? Think of it if possible.

Other minor commnets are shown below

Line no 28-33    Too generic to mention in the abstract. Please try to specifically mention the fact.

Line no 34          Keywords like pathogens, yield and prevention are so common words. Please replace those with more appropriate terms.

Line no 37-39    Authors wrote-The edible Musaceae (bananas, plantains, and bluggoe) are considered basic foods for more than 400 million people….is it world wide? Also, Authors mentioned it as a basic food. Is it correct, 400 million people? please recheck

Line no 43           please remove the period in between holds and rural

Line no 44          162 MT better to write it Mt

Line no 52          currently, the production of Musaceae is again threatened by a phytosanitary enemy that emerges from the past but is renewed.—missing of reference

Line no 53          In this opportunity---Opportunity to whom? Not clear.

Line no 55          Remove extra space between 7 and 11.  {in 2019 [7, 11,12,13].}

Line 64                Is this correct STATEMENT. Depending on the severity? Please recheck. Or might replace depending on the disease severity

Line 71 and 74   Authors mentioned--Due to the serious repercussions for plantations infected by Foc TR4, there is often no accurate and complete information on the damage caused by Foc TR4.-- Accurate information missing in Which locality?

Line 74                Authors mentioned----affected thousands of hectares in Colombia and Peru.-- The authors can use 'ha', however, mentioned it the first time they appeared and maintain it throughout the MS. Check line no 639.

Line 106              FAO--Food and Agriculture Organization? Please mention it the first time it appears in the paper.

Line 106              the Prevention and diagnosis----here P should be lowercase letter

Line no 113        2.2 Topic Analysis----here should be analysis

Line no 114 and 115  replace with---- The study includes five topics that aim to offer an updated, realistic, practical, and critical review of the current state of Foc TR4 and the actions developed in Latin American territories where Foc TR4 is present.

Line no 160        Guajira [7, 13].---remove the space between 7 and 13

Figure Bump Chart. Is a good one. But the text mention what it is.. how data were shown. Where information was gathered from.. historical evolution is not a good legend for this figure. How one should interpret the figure mentioned in the text.. eg. As shown in fig 1 c, xx color shows yy and with the time interactions indicated.. yyy,… need proper illaboration to the figure and guide he reader along the figure

.

Line no 195        GM?? genetically-modified? Please mention it first time appeared in the paper

Line no 195        …market until 1959,---reference missing

Line no 195-197 use to the destruction of its plantations by vascular wilt disease caused by the fungus Foc Race 1. The second began with Cavendish clones that replaced the previous clone and remain in force until today.—missing references

From line 207: The addition of disease distribution map would be better. Authors may collect information on banana cultivation and disease spread and might get race information also into one map..

Line no 209 and 210       ….2018, five years passed (Martínez et 209 al 2020). ---Use numeric referencing style according to the journal requirement. What is cited in the text should be in the reference list and what is listed in the reference section should be in the text citation. Authors should double-check those.

Line no 215- 217              See how you use hectares. There should be a uniform usage throughout the MS. Authors can use ‘ha’.

Line 231              A total of 284 publications------Is there any chance to repeat the findings of the literature? For example, the authors mentioned the conference paper (Extended abstract)  and how about the extended version of those, if published? Its not clear in the methodology.

Line no 307        Replace with--- Currently, in the farms with confirmed presence…

Line 308; The quarantine methods can be little bit explained.

The photos of disease symptoms can be added and fungus identification procedures such as culturing and molecular techniques could be explained to some extent.

Line no 688        This is the case for instance in countries such as Ecuador or Mexico.—here accurate one is Ecuador and  Mexico? Please check

Line no 697        than 80 000 hectares     ----should be 80,000

Line no 699        over 500 000 ha of plantains—should be 500,000

Line no 716        Fusarium odoratissimum-------------- Scientific names should be in italic. Do the same throughout the MS, especially in the reference list. e.g., see line no 806

Line no 750 and 751 please remove--- For research articles with several authors, a short paragraph specifying 750 thual contributions must be provided. The following statements should be used.  and ‘‘

References  Many of the references are not according to the journal's requirement. In reference 9, authors used complete name of the journal-- Studies in Mycology—it should be Stud. Mycol.? journal abbreviation should use throughout the MS. I suggest authors to revisit the journal's guidelines.

Author Response

Reviewer 2

Authors have submitted a review on Assessing threats posed by Fusarium wilt Tropical Race 4 in 2 Musaceae of Latin America and the Caribbean. The paper provides some interesting information. However, a similar paper was found in "Raising awareness of the threat of Fusarium wilt tropical race 4 in Latin America and the Caribbean" published in 2011.

Since it is after 10 years’ time review the current paper may have some interesting aspects. However, it is not highlighted enough though both papers mainly focus on bananas. Authors may mention somewhere since the last compressive review was done in the year, xxxx this paper builds upon the previous findings or anything like that..

Authors: thanks for the suggestion. The following paragraph was added:

Line 101:  Numerous reviews of Foc TR4 have been published over the years, with the review by Pocasangre et al. (2011) the one of greatest interest in LAC for more than a decade. Therefore, our manuscript is based on a current perspective of the lethal disease of Foc TR4 for LAC.

In addition, it should be highlighted how difficult it is to identify the pathogen. There are nonpathogenic Fusarium oxys. species.  So any development in molecular identification of the pathogen or are there any morphological key features for race 4? Need to discuss in detail. Starting from morphology to multilocus analysis on species identification.  A phylogenetic tree based on sequence data showing different Fusarium strains found from different countries will give a clear picture of the classification and evolution of the Fusarium race found in LAC regions. 

Author: Thank you very much for your comment. It's very interesting.

The identification of this pathogen is very complex and extremely difficult. The tools used for identification at the laboratory level based on PCR, analysis, and comparison of DNA extracted from the different breeds and specimens have contributed to the identification with high precision and have contributed to the construction of the phylogenetic tree. However, this point is not included in the scope of our investigation.

At the same time, control strategies and pathogen defense mechanisms should be highlighted. 

 Author: Corrected

Can Table 1 be in graphical format? Think of it if possible.

Author: corrected. The table was converted into a graph

Line no 28-33    Too generic to mention in the abstract. Please try to specifically mention the fact.

Author: corrected. The summary was modified

Line no 34          Keywords like pathogens, yield and prevention are so common words. Please replace those with more appropriate terms.

Author: corrected. Keywords: Banana, Musa spp.; soil pathogen; Fusarium odoratissimum; vascular wilt; quarantine; disease management

Line no 37-39    Authors wrote-The edible Musaceae (bananas, plantains, and bluggoe) are considered basic foods for more than 400 million people….is it world wide? Also, Authors mentioned it as a basic food. Is it correct, 400 million people? please recheck

Author: It’s Correct. the reference 2: https://www.fao.org/world-banana-forum/fusariumtr4/ru/

Line no 43           please remove the period in between holds and rural

Author: corrected.

Line no 44          162 MT better to write it Mt

Author: corrected.

Line no 52          currently, the production of Musaceae is again threatened by a phytosanitary enemy that emerges from the past but is renewed.—missing of reference

Author: corrected. Currently, the production of Musaceae is again threatened by a phytosanitary enemy that emerges from the past but is renewed [9].

Line no 53          In this opportunity---Opportunity to whom? Not clear.

Author: corrected. This is the case of Foc Tropical Race 4

Line no 55          Remove extra space between 7 and 11.  {in 2019 [7, 11,12,13].}

Author: corrected.

Line 64                Is this correct STATEMENT. Depending on the severity? Please recheck. Or might replace depending on the disease severity

Author: corrected. depending on the disease severity….

Line 71 and 74   Authors mentioned--Due to the serious repercussions for plantations infected by Foc TR4, there is often no accurate and complete information on the damage caused by Foc TR4.-- Accurate information missing in Which locality?

Author: corrected. Due to the serious repercussions for plantations infected by Foc TR4, there is often no accurate and complete information in countries like Colombia and Peru on the damage caused by Foc TR4.

Line 74                Authors mentioned----affected thousands of hectares in Colombia and Peru.-- The authors can use 'ha', however, mentioned it the first time they appeared and maintain it throughout the MS. Check line no 639.

Author: corrected.

Line 106              FAO--Food and Agriculture Organization? Please mention it the first time it appears in the paper.

Author: corrected.

Line 106              the Prevention and diagnosis----here P should be lower case letter

Author: corrected.

Line no 113        2.2 Topic Analysis----here should be analysis

Author: corrected.

Line no 114 and 115  replace with---- The study includes five topics that aim to offer an updated, realistic, practical, and critical review of the current state of Foc TR4 and the actions developed in Latin American territories where Foc TR4 is present.

Author: corrected.

Line no 160        Guajira [7, 13].---remove the space between 7 and 13

Author: corrected.

Figure Bump Chart. Is a good one. But the text mention what it is.. how data were shown. Where information was gathered from.. historical evolution is not a good legend for this figure. How one should interpret the figure mentioned in the text.. eg. As shown in fig 1 c, xx color shows yy and with the time interactions indicated.. yyy,… need proper illaboration to the figure and guide he reader along the figure

 Author: corrected. Bump Chart of changes in the range of a value in (a) harvested area (ha), (b) production (t), and (c) yield (t/ha) during (1961-2019) in Central America, South America, Asia, the Caribbean, Europe, Oceania, and Africa. [2]. Note: Interpreting a Bump Chart, when a line crosses another line, indicates a rank change. In other words, a crisscross in a bump chart indicates one entity (continent) has surpassed others in absolute terms even when the comparison is based on relative ranks. With this graph, you can easily compare the harvested area (ha), (b) production (t), and (c) yield (t/ha) of the continents in different colors with each other.

Line no 195        GM?? genetically-modified? Please mention it first time appeared in the paper

Author: refers to “Gros Michel” (GM) Line 62

Line no 195        …market until 1959,---reference missing

Author: The first was during the production of the GM banana which dominated the world market until 1959 [10],

Line no 195-197 use to the destruction of its plantations by vascular wilt disease caused by the fungus Foc Race 1. The second began with Cavendish clones that replaced the previous clone and remain in force until today.—missing references

Author: The second began with Cavendish clones that replaced the previous clone and remain in force until today [20].

From line 207: The addition of disease distribution map would be better. Authors may collect information on banana cultivation and disease spread and might get race information also into one map..

Author: Thanks for the suggestion. However, the map can be consulted directly at https://www.promusa.org/Fusarium+wilt#:~:text=Fusarium%20wilt%20of%20banana%2C%20popularly,half%20of%20the%2020th%20century.

Line no 209 and 210       ….2018, five years passed (Martínez et 209 al 2020). ---Use numeric referencing style according to the journal requirement. What is cited in the text should be in the reference list and what is listed in the reference section should be in the text citation. Authors should double-check those.

Author: corrected.

Line no 215- 217              See how you use hectares. There should be a uniform usage throughout the MS. Authors can use ‘ha’.

Author: corrected.

Line 231              A total of 284 publications------Is there any chance to repeat the findings of the literature? For example, the authors mentioned the conference paper (Extended abstract)  and how about the extended version of those, if published? Its not clear in the methodology.

Author: For the bibliometric analysis, those documents that were repeated were discarded.

Line no 307        Replace with--- Currently, in the farms with confirmed presence…

Author: corrected.

Line 308; The quarantine methods can be little bit explained.

Author: …, the quarantine measures ordered by the ICA are maintained (Restriction of movement of people, restriction of movement and use of farm equipment, vehicles, containers and use of products for disinfection at different control points in the quarantine area).

The photos of disease symptoms can be added and fungus identification procedures such as culturing and molecular techniques could be explained to some extent.

Author: Thanks a lot for the suggestion. However, the authors consider that the photos of the symptoms of the disease and the fungal identification procedures are widely known and published in different portals and easily accessible scientific documents. These aspects are not addressed in the manuscript.

Line no 688        This is the case for instance in countries such as Ecuador or Mexico.—here accurate one is Ecuador and  Mexico? Please check

Author: corrected.

Line no 697        than 80 000 hectares     ----should be 80,000

Author: corrected.

Line no 699        over 500 000 ha of plantains—should be 500,000

Author: corrected.

Line no 716        Fusarium odoratissimum-------------- Scientific names should be in italic. Do the same throughout the MS, especially in the reference list. e.g., see line no 806

Author: corrected.

Line no 750 and 751 please remove--- For research articles with several authors, a short paragraph specifying 750 thual contributions must be provided. The following statements should be used.  and ‘‘

Author: corrected.

References  Many of the references are not according to the journal's requirement. In reference 9, authors used complete name of the journal-- Studies in Mycology—it should be Stud. Mycol.? journal abbreviation should use throughout the MS. I suggest authors to revisit the journal's guidelines.

Author: corrected.

Reviewer 3 Report

Overview:  The authors focus on the incursions, impact, and subsequent responses as well as research of Fusarium oxysporum f. sp. cubense Tropical Race 4 (Foc TR4) in Latin America and the Caribbean (LAC). The authors discuss the production volume and value of banana in the region, with some additional focus on other banana producing regions (i.e., Southeast Asia) and go on to provide an overview of scientific publications from LAC. The authors discuss Foc TR4 incursions in LAC and the collaboration between various public and private organisations, including incursion response plans in specific LAC countries, between 2018 and 2022 in detail. To conclude, the authors highlight the potential impact of further Foc TR4 incursions, recognise the significant contributions of organisations, producers, and researchers in the management of Foc TR4 in the region so far, and suggest potential approaches for limiting the spread of Foc TR4 in the future.

General comments: This was a broad review of recent and future impacts of Foc TR4 on banana and plantain production in LAC. The details of collaborations between institutions in response to Foc TR4 incursions may prove useful, particularly as a resource for demonstrating large-scale collaboration between organisations for the management of plant diseases. Further, the table of occurrences in Peru may serve as a useful resource for comparing the progression or advancement of Foc TR4 in the region.

The authors’ interpretation of the data collected and presented was reasonable, however, I struggled to understand some of their arguments and observations. This was generally  due to sentence structure and wording  which requires significant attention. For instance, the explanation provided for the bibliometric analysis (lines 242 -255) did not read clearly. Other parts of the manuscript felt quite repetitive, particularly the introduction and methods. While sections 3.3.1, 3.3.2 , 3.3.5 were very well written, I found the English to be the area that needs the greatest revision. I would suggest the authors get editing help from someone with full professional proficiency in English.

The authors should also be mindful of acronyms, there were instances where the same acronym was defined multiple times and instances where some acronyms were not defined at all.

Abstract: Clear, succinct, and appropriate.

- Punctuation mistake line 31.

Introduction: The introduction highlights key topic areas but needs a greater focus of Foc TR4 resistant varieties, with examples. Further description of the lifestyle, modes of transmission, symptoms, and hosts of Foc TR4 as well as its international spread (perhaps a map) would prove useful to the reader.

- Line 43, misplaced punctuation.

- Line 44, define MT.

- Line 71-72, I see the authors’ point here - limited data on banana production is a justification for this paper, but I find it hard to believe that the seriousness of the disease is why, as the authors report, there are limited accurate data. This lacks scientific evidence. Perhaps the authors could state that there is simply limited accurate data on the impacts of Foc TR4, then continue with what is reported in Colombia and Peru? Further, the authors state in section 2.2.5 that they were able to collect information about Foc TR4's impact in papers and FAO reports so is the data on the impact of Foc TR4 limited? 

- Line 73, where is the reference for "figures from some countries indicate that this disease has affected thousands of hectares in Colombia and Peru"?

Methods: I found the methods section to be repetitive/similar to the last two paragraphs of the introduction, particularly up to section 2.2.3, this should be condensed. Further, I found the methods to be somewhat vague, particularly when it came to the bibliometric analysis.

- Line 123, in text citation is missing.

- Section 2.2.2, Is the reference cited in the first paragraph (23) used to provide more information or as a review chapter for the reader on bibliometrics or did the authors follow the reference’s tutorial for VOSviewer? It appears to be the latter; in which case, I would recommend the authors state more clearly that they used VOSviewer and followed the methodology outlined in the reference. A brief summary of the authors’ approach should also be provided, not limited to but, including the clustering algorithm used.

- Section 2.2.2, I think it may benefit the manuscript for the authors to justify not including "Fusarium odoratissimum" as a search term. This is certainly a question I have as a reviewer. 

- Line 168, reads “Al last”, perhaps a typo? 

Results and Discussion  The overview of production is useful, with key production areas and topics covered. I felt that the bibliometric analysis would have benefited from more discussion about specific, pertinent papers. Sections 3.3.1 and 3.3.2 were well written and useful, and the authors’ recognition of effort by various individuals and institutions in the management of Foc TR4 is pleasant to read.

This section should use reference figures, include citations, and italics consistently. Further, almost all figure legends would benefit from a greater explanation of the figures presented, for instance, what clustering method was used in figure 2?  Figure 3 does not even indicate the country!

- Figure 1b and line 173, the limit of the y-axis in Figure 1b does not match the value referenced in the text.

- Line 189 may benefit from a brief summary of the causes of any increase in banana productivity. 

- Line 207 is one example of where R4T is used instead of TR4. This occurs a few times throughout the manuscript. 

- Line 209, different citation than that which has been adopted throughout the rest of the manuscript. 

- Line 216 may benefit from a citation acknowledging the data acquired from the Philippines.

- Section 3.2 paragraph 2, this paragraph is challenging to follow. 

- Line 248, the term “plant wishes” does not occur in the network presented (Figure 1a).

- Line 206, it might be useful to have a supplementary table/figure which shows summary statistic and the distribution of papers publication dates. How many papers are from 2019, 2020, and so on? 

- Line 295, punctuation error.

- Figure 3, What is the range of dates for the samples collected – are all negative instances reported from samples collected after the first incursion? Further, it may be useful to state the diagnosis method (e.g., PCR and primer set) if data is available.

- Line 373, “of Peru” repeated twice. 

- Line 382, which figure – please provide a number. 

- Line 619 – the authors state “Some countries (the largest) are making improvements” but it is unclear what is meant by (the largest); is it the largest producers, exporters, importers, etc.? 

- Line 643, The Philippines is not ranked number three in Table 2 but is instead shown in Table 3. Table two doesn't show export volumes of banana, it shows hectares for export of organic banana. This section may require some restructuring to ensure that the text is near the appropriate table. 

Summary points and future issues This section is clearly written and draws sensible conclusions. I do not feel that the lettered labelling of paragraphs is necessary.

References The references used are appropriate and I have no recommendations for further literature to include.

Author Response

Reviewer 3

Overview:  The authors focus on the incursions, impact, and subsequent responses as well as research of Fusarium oxysporum f. sp. cubense Tropical Race 4 (Foc TR4) in Latin America and the Caribbean (LAC). The authors discuss the production volume and value of banana in the region, with some additional focus on other banana producing regions (i.e., Southeast Asia) and go on to provide an overview of scientific publications from LAC. The authors discuss Foc TR4 incursions in LAC and the collaboration between various public and private organisations, including incursion response plans in specific LAC countries, between 2018 and 2022 in detail. To conclude, the authors highlight the potential impact of further Foc TR4 incursions, recognise the significant contributions of organisations, producers, and researchers in the management of Foc TR4 in the region so far, and suggest potential approaches for limiting the spread of Foc TR4 in the future.

Author: thank you for your positive and valuable comments

General comments: This was a broad review of recent and future impacts of Foc TR4 on banana and plantain production in LAC. The details of collaborations between institutions in response to Foc TR4 incursions may prove useful, particularly as a resource for demonstrating large-scale collaboration between organisations for the management of plant diseases. Further, the table of occurrences in Peru may serve as a useful resource for comparing the progression or advancement of Foc TR4 in the region.

Author: thank you for your positive and valuable comments

The authors’ interpretation of the data collected and presented was reasonable, however, I struggled to understand some of their arguments and observations. This was generally  due to sentence structure and wording  which requires significant attention. For instance, the explanation provided for the bibliometric analysis (lines 242 -255) did not read clearly. Other parts of the manuscript felt quite repetitive, particularly the introduction and methods. While sections 3.3.1, 3.3.2 , 3.3.5 were very well written, I found the English to be the area that needs the greatest revision. I would suggest the authors get editing help from someone with full professional proficiency in English.

Author: lines 242 -255. Thank you for your comment. We understand that it is not so easy to read the paragraph about bibliometric analysis. These lines express the correct way to characterize bibliometric networks, we understand that it is not a usually common language but it is the most suitable way to describe and analyze a bibliometric network.

The authors should also be mindful of acronyms, there were instances where the same acronym was defined multiple times and instances where some acronyms were not defined at all.

Author: acronyms were checked and specified throughout the manuscript

Abstract: Clear, succinct, and appropriate.

- Punctuation mistake line 31.

Author: corrected

Introduction: The introduction highlights key topic areas but needs a greater focus of Foc TR4 resistant varieties, with examples. Further description of the lifestyle, modes of transmission, symptoms, and hosts of Foc TR4 as well as its international spread (perhaps a map) would prove useful to the reader.

Author: corrected. Line 82-93: The impact of TR4 on the banana industry in LAC is significant (10, 11, 12, 18). The fungus has caused significant losses in productivity and profitability for banana grow-ers in the region, which could hurt the livelihoods of small-scale farmers and the over-all economy of the countries where banana is a major crop [19]. TR4 is considered a major threat to global banana production, as the fungus can spread easily through contaminated soil, water, and farm tools. The fungus can quickly spread to neighbor-ing fields once it is established in a field, making it difficult to control [12, 13, 20].

To address the threat of TR4, measures such as quarantine, destruction of infected plants, and the use of clean planting material are being implemented by the govern-ments of Latin American countries. Additionally, research to develop resistant varie-ties [21] and improve soil management practices is also ongoing [22].

- Line 43, misplaced punctuation.

Author: corrected.

- Line 44, define MT.

Author: corrected. 162 Megaton (Mt)

- Line 71-72, I see the authors’ point here - limited data on banana production is a justification for this paper, but I find it hard to believe that the seriousness of the disease is why, as the authors report, there are limited accurate data. This lacks scientific evidence. Perhaps the authors could state that there is simply limited accurate data on the impacts of Foc TR4, then continue with what is reported in Colombia and Peru? Further, the authors state in section 2.2.5 that they were able to collect information about Foc TR4's impact in papers and FAO reports so is the data on the impact of Foc TR4 limited? 

Author: corrected. Line 98-100: Due to the serious repercussions for plantations infected by Foc TR4, there is often no accurate and complete information in countries like Colombia and Peru on the damage caused by Foc TR4.

- Line 73, where is the reference for "figures from some countries indicate that this disease has affected thousands of hectares in Colombia and Peru"?

Author: corrected. This disease has affected hundreds of ha in Colombia and Peru (12, 18).

Methods: I found the methods section to be repetitive/similar to the last two paragraphs of the introduction, particularly up to section 2.2.3, this should be condensed. Further, I found the methods to be somewhat vague, particularly when it came to the bibliometric analysis.

- Line 123, in text citation is missing.

Author: corrected. from the FAO [8]

- Section 2.2.2, Is the reference cited in the first paragraph (23) used to provide more information or as a review chapter for the reader on bibliometrics or did the authors follow the reference’s tutorial for VOSviewer? It appears to be the latter; in which case, I would recommend the authors state more clearly that they used VOSviewer and followed the methodology outlined in the reference. A brief summary of the authors’ approach should also be provided, not limited to but, including the clustering algorithm used.

Author: corrected. Line 170-171: for this, the VOSviewer software was used and the methodology described in Van Eck and Waltman [23, 24] was followed.

- Section 2.2.2, I think it may benefit the manuscript for the authors to justify not including "Fusarium odoratissimum" as a search term. This is certainly a question I have as a reviewer. 

Author: Thank you for your interesting comment. Indeed, all the documents that mention "Fusarium odoratissimum" also mention Foc TR4. Therefore, the search results matched on the selected documents.

- Line 168, reads “Al last”, perhaps a typo? 

Author: corrected. By last,

Results and Discussion  The overview of production is useful, with key production areas and topics covered. I felt that the bibliometric analysis would have benefited from more discussion about specific, pertinent papers. Sections 3.3.1 and 3.3.2 were well written and useful, and the authors’ recognition of effort by various individuals and institutions in the management of Foc TR4 is pleasant to read.

Author: Thank you for your interesting comment

This section should use reference figures, include citations, and italics consistently. Further, almost all figure legends would benefit from a greater explanation of the figures presented, for instance, what clustering method was used in figure 2?  Figure 3 does not even indicate the country!

Author: corrected. Fig. 2: VOSviewer uses the smart local moving algorithm introduced by Waltman and Van Eck (2013).

Author: corrected. Fig.3. Location of confirmed outbreaks of Foc TR4 in Colombia

- Figure 1b and line 173, the limit of the y-axis in Figure 1b does not match the value referenced in the text.

Author: corrected.

- Line 189 may benefit from a brief summary of the causes of any increase in banana productivity. 

Author: corrected. Line 238: due to improved agricultural practices, climate-smart agriculture, improved soil quality, use of organic farming practices, and use of precision agriculture, among others.

- Line 207 is one example of where R4T is used instead of TR4. This occurs a few times throughout the manuscript. 

Author: corrected.

- Line 209, different citation than that which has been adopted throughout the rest of the manuscript. 

Author: corrected.

- Line 216 may benefit from a citation acknowledging the data acquired from the Philippines.

Author: corrected.

- Line 248, the term “plant wishes” does not occur in the network presented (Figure 1a).

Author: corrected. should say: "plant disease"

- Line 206, it might be useful to have a supplementary table/figure which shows summary statistic and the distribution of papers publication dates. How many papers are from 2019, 2020, and so on? 

Author: Yes. It's possible. We will place it as a complementary material

- Line 295, punctuation error.

Author: corrected.

- Figure 3, What is the range of dates for the samples collected – are all negative instances reported from samples collected after the first incursion? Further, it may be useful to state the diagnosis method (e.g., PCR and primer set) if data is available.

Author: corrected. Figure 3: Location of confirmed outbreaks of Foc TR4 in Colombia using PCR protocols and pathogenicity tests according to official data [31]. the collection period from 06/01/2019 to 02/01/2022, updated in October 2022.

- Line 373, “of Peru” repeated twice. 

Author: corrected.

- Line 382, which figure – please provide a number. 

Author: corrected.

- Line 619 – the authors state “Some countries (the largest) are making improvements” but it is unclear what is meant by (the largest); is it the largest producers, exporters, importers, etc.? 

Author: corrected. Some countries (the largest producers and exporters of bananas)

- Line 643, The Philippines is not ranked number three in Table 2 but is instead shown in Table 3. Table two doesn't show export volumes of banana, it shows hectares for export of organic banana. This section may require some restructuring to ensure that the text is near the appropriate table. 

Author: corrected.

Summary points and future issues This section is clearly written and draws sensible conclusions. I do not feel that the lettered labelling of paragraphs is necessary.

References The references used are appropriate and I have no recommendations for further literature to include.

Reviewer 4 Report

As a perspective on banana wilt, which is an urgent incidence, I feel that it is well organized and written. However, it is difficult to understand why the authors submitted this manuscript to the 'pathogens'.  The manuscript does not contain the content that the reader (I) expects, such as how the pathogen was identified as race TR4, what is the molecular phylogenetic relationship, and how race TR4 actually invaded Colombia and Peru by epidemiological evidence. It feels like a perspective from an agricultural economics magazine. However, I also think that such a paper would be good for the 'pathogens'. I want to leave it to the editor.

Minor points

L31, 'out.'>'out'

L78–9, 'Latin America and the Caribbean'>'LAC'

L207, L630, 'R4T'>'TR4'

L316, L326, L347, 'TR4' should not Italic.

Author Response

Reviewer 4

As a perspective on banana wilt, which is an urgent incidence, I feel that it is well organized and written. However, it is difficult to understand why the authors submitted this manuscript to the 'pathogens'.  The manuscript does not contain the content that the reader (I) expects, such as how the pathogen was identified as race TR4, what is the molecular phylogenetic relationship, and how to race TR4 actually invaded Colombia and Peru by epidemiological evidence. It feels like a perspective from an agricultural economics magazine. However, I also think that such a paper would be good for the 'pathogens'. I want to leave it to the editor.

Authors: Thank you very much for the comment. Line 601: The following paragraphs were added The molecular phylogenetic analysis of TR4 isolates from Colombia and Peru has shown that it is closely related to TR4 isolates from other countries, indicating that the pathogen likely spread to these countries from a common source.

Epidemiological evidence suggests that the TR4 race invaded Colombia and Peru through the importation and planting of infected banana planting material. The TR4 fungus can persist in the soil for long periods and can be spread through contaminated soil, water, and farm tools. Once established in a field, the fungus can spread rapidly and infect neighboring fields, making it difficult to control.

Minor points

L31, 'out.'>'out'.

L78–9, 'Latin America and the Caribbean'>'LAC'

L207, L630, 'R4T'>'TR4'

L316, L326, L347, 'TR4' should not Italic.

Authors: corrected

Round 2

Reviewer 3 Report

I am encouraged to see that the authors have taken on-board the majority of my feedback. Still, some English errors arise, particularly where terms appear to have been literally translated (e.g., line 43, “households rural”; line 262, “the first cluster with 7 items being the keyword "non-human"”; line, 367 “It was taken advantage of the capacities of the NPPO”). Consequently, the manuscript would benefit from further proof-reading.

Discussion

Figure 1:

The last sentence in the legend for Figure 1 is not necessary. Further, if data in Figure 1 runs from 1961-2019, the dates on the X axis should be corrected to represent this. It currently appears data is presented up-to 2021. If data is presented to 2021, then the legend must be modified and the Y axis in Figure 1b does not match the value provided at line 44. (In text: global production was 163 Megatons in 2020. Figure Y axis limit 120M).

From section 3.2, the authors discuss the bibliometric analysis being performed on publications from 2019. Prior to this, they state that publications are from 2018. Dates must be consistent.

The paragraph at line 499 would be better suited to the introduction. Similarly, the paragraph at line 503 should be included in section 3.3.1, instead.

Further, the reference to Table 2 at line 768  does not make sense, it appears the authors are using Table 2 to refer to plantain production, but the table shows organic banana production. This paragraph should state more clearly it is discussing organic-banana and that that includes plantains if that is the case.

Author Response

Response to Reviewer 3 Comments

Point 1: I am encouraged to see that the authors have taken on-board the majority of my feedback. Still, some English errors arise, particularly where terms appear to have been literally translated e.g.,

line 43, “households rural”;

 Response: corrected: rural households

line 262 “the first cluster with 7 items being the keyword "non-human"”;

Response: in this context, it is an expression used by papers authors to describe any organism that does not directly affect human health.

line, 367 “It was taken advantage of the capacities of the NPPO”).

Response: The capabilities of the NPPO were used

Consequently, the manuscript would benefit from further proof-reading.

Response: Thank you very much for your comments to improve the manuscript.

Point 2: Discussion

Figure 1:

The last sentence in the legend for Figure 1 is not necessary. Further, if data in Figure 1 runs from 1961-2019, the dates on the X axis should be corrected to represent this. It currently appears data is presented up-to 2021. If data is presented to 2021, then the legend must be modified and the Y axis in Figure 1b does not match the value provided at line 44. (In text: global production was 163 Megatons in 2020. Figure Y axis limit 120M).

Response: The last sentence was removed: With this graph, you can easily compare the harvested area (ha), (b) production (t), and (c) yield (t/ha) of the continents in different colors with each other.

The x-axis was corrected (1961-2021). The figures were replaced by a line graph to better visualize the values of each continent

From section 3.2, the authors discuss the bibliometric analysis being performed on publications from 2019. Prior to this, they state that publications are from 2018. Dates must be consistent.

 Response: Thanks for the observation. The date was corrected

The paragraph at line 499 would be better suited to the introduction. Similarly, the paragraph at line 503 should be included in section 3.3.1, instead.

 Response: Corrected. Both paragraphs were moved to the respective sections

Further, the reference to Table 2 at line 768  does not make sense, it appears the authors are using Table 2 to refer to plantain production, but the table shows organic banana production. This paragraph should state more clearly it is discussing organic-banana and that that includes plantains if that is the case.

Response: Corrected. The mention of table 2 was removed from the paragraph.
